# UNSUPERVISED OBJECT-ORIENTED 3D SCENE REPRESENTATION AND RENDERING

## ABSTRACT

In this paper, we propose a probabilistic generative model, called ROOTS, for unsupervised learning of object-oriented 3D-scene representation and rendering. ROOTS bases on the Generative Query Network (GQN) framework. However, unlike GQN, ROOTS provides independent, modular, and object-oriented decomposition of the 3D scene representation. In ROOTS, the inferred object-oriented representation is 3D in the sense that it is 3D-viewpoint invariant as the scene-level representation of GQN is so. ROOTS also provides hierarchical object-oriented representation: at 3D global-scene level and at 2D local-image level. In experiments, we demonstrate on datasets of 3D rooms with multiple objects, the above properties by focusing on its abilities for disentanglement, compositionality, transferability, and generalization. ROOTS achieves this without performance degradation on generation quality in comparison to GQN.

## 1 INTRODUCTION

The shortcomings of contemporary deep learning such as interpretability, sample efficiency, ability for reasoning and causal inference, transferability, and compositionality, are where the symbolic AI has traditionally shown its strengths (Garnelo & Shanahan, 2019). Thus, one of the grand challenges in machine learning has been to make deep learning embrace the benefits of symbolic representation so that symbolic entities can emerge from high-dimensional observations such as visual scenes.

In particular, for learning from visual observations of the physical world, such representation should consider the following criteria. First, it should focus on *objects* (and their *relations*) which are foundational entities constructing the physical world. These can be considered as units on which we can build a modular model. The modular nature also helps compositionality (Andreas et al., 2016) and transferability (Kansky et al., 2017). Second, being *three-dimensional* (3D) is a decisive property of the physical world. We humans, equipped with such 3D representation in our brain (Yamane et al., 2008), can retain consistency on the identity of an object even if it is observed from different viewpoints. Lastly, learning such representation should be *unsupervised*. Although there have been remarkable advances in supervised methods to object perception (Redmon et al., 2016; Ren et al., 2015; Long et al., 2015), the technology should advance toward unsupervised learning as we humans do. This not only avoids expensive labeling efforts but also allows adaptability and flexibility to the evolving goals of various downstream tasks because "objectness" itself can vary on the situation.

In this paper, we propose a probabilistic generative model that can learn, without supervision, object-oriented 3D representation of a 3D scene from its partial 2D observations. We call the proposed model ROOTS (Representation of Object-Oriented Three-dimensional Scenes). We base our model on the framework of Generative Query Networks (GQN) (Eslami et al., 2018; Kumar et al., 2018). However, unlike GQN which provides only a scene-level representation that encodes the whole 3D scene into a single continuous vector, the scene representation of ROOTS is decomposed into object-wise representations each of which is also an independent, modular, and 3D representation. Further, ROOTS learns to model a background representation separately for the non-object part of the scene. The object-oriented representation of ROOTS is more interpretable, composable, and transferable. Besides, ROOTS provides the two-level *hierarchy* of the object-oriented representation: one for a global 3D scene and another for local 2D images. This makes the model more interpretable and provides more useful structure for downstream tasks. In experiments, we show the above abilities of ROOTS on the 3D-Room dataset containing images of 3D rooms with several objects of different

colors and shapes. We also show that these new abilities are achieved without sacrificing generation quality compared to GQN.

Our proposed problem and method are significantly different from existing works on visual 3D learning although some of those partly tackle some of our challenges. First, our model learns factorized object-oriented 3D representations which are independent and modular, from a scene containing multiple objects with occlusion and partial observability rather than a single object. Second, our method is unsupervised, not using any 3D structure annotation such as voxels, cloud points, or meshes as well as bounding boxes or segmentation annotation. Third, our model is a probabilistic generative model learning both representation and rendering with uncertainty modeling. Lastly, it is trained end-to-end. In Section 4, we provide more discussion on the related works.

The main contributions are: (i) We propose, in the GQN framework, a new problem of learning object-oriented 3D representations of a 3D scene containing multiple objects with occlusion and partial observability in the challenging setting described above. (ii) We achieve this by proposing a new probabilistic model and neural architecture. (iii) We demonstrate that our model enables various new abilities such as compositionality and transferability while not losing generation quality.

## 2 PRELIMINARY: GENERATIVE QUERY NETWORKS

The generative query networks (GQN) (Eslami et al., 2018) is a probabilistic generative latent-variable model providing a framework to learn a 3D representation of a 3D scene. In this framework, an agent navigating a scene $i$ collects $K$ images $\mathbf{x}_i^k$ from 2D viewpoint $\mathbf{v}_i^k$. We refer this collection to context observations $C_i = \{(\mathbf{x}_i^k, \mathbf{v}_i^k)\}_{k=1,\dots,K}$. While GQN is trained on a set of scenes, in the following, we omit the scene index $i$ for brevity and discuss a single scene without loss of generality. GQN learns scene representation $\mathbf{z}$ from context $C$. The learned representation $\mathbf{z}$ of GQN is a *3D-viewpoint invariant* representation of the scene in the sense that, given an arbitrary query viewpoint $\mathbf{v}^q$, its corresponding 2D image $\mathbf{x}^q$ can be generated from the representation.

In the GQN framework, there are two versions. The standard GQN model (Eslami et al., 2018) uses the query viewpoint to generate representation whereas the Consistent GQN (CGQN) (Kumar et al., 2018) uses the query *after* generating the scene representation in order to obtain query-independent scene-level representation. Although we use CGQN as our base framework to obtain query-independent scene-level representation, in the rest of the paper we use the abbreviation GQN instead of CGQN to indicate the general GQN framework embracing both GQN and CGQN.

The generative process of GQN is written as follows: $p(\mathbf{x}^q|\mathbf{v}^q, C) = \int p(\mathbf{x}^q|\mathbf{v}^q, \mathbf{z})p(\mathbf{z}|C)\mathrm{d}\mathbf{z}$. As shown, GQN uses a conditional prior $p(\mathbf{z}|C)$ to learn scene representation $\mathbf{z}$ from context. To do this, it first obtains a neural scene representation $\mathbf{r}$ from the representation network $\mathbf{r} = f_{\text{repr-gqn}}(C)$ which combines the encodings of $(\mathbf{v}^k, \mathbf{x}^k) \in C$ in an order-invariant way such as sum or mean. It then uses ConvDRAW (Gregor et al., 2016) to generate the scene latent variable $\mathbf{z}$ from scene representation $\mathbf{r}$ by $p(\mathbf{z}|C) = \prod_{l=1:L} p(\mathbf{z}^l|\mathbf{z}^{<l}, \mathbf{r}) = \text{ConvDRAW}(\mathbf{r})$ with $L$ autoregressive rollout steps. Due to intractability of the posterior distribution $p(\mathbf{z}|C, \mathbf{v}^q, \mathbf{x}^q)$, GQN uses variational inference for posterior approximation and the reparameterization trick (Kingma & Welling, 2013) for backpropagation through stochastic variables. The objective is to maximize the following evidence lower bound (ELBO) via gradient-based optimization.

$$\log p_\theta(\mathbf{x}^q \mid \mathbf{v}^q, C) \geq \mathbb{E}_{q_\phi(\mathbf{z}|C, \mathbf{v}^q, \mathbf{x}^q)}\left[\log p_\theta(\mathbf{x}^q \mid \mathbf{v}^q, \mathbf{z})\right] - \mathbb{KL}(q_\phi(\mathbf{z} \mid C, \mathbf{v}^q, \mathbf{x}^q) \parallel p_\theta(\mathbf{z} \mid C)).$$

Note that although in this paper we use a single target observation $D = (\mathbf{x}^q, \mathbf{v}^q)$ for brevity, the model is in general trained on a set of target observations $D = \{(\mathbf{x}_j^q, \mathbf{v}_j^q)\}_j$.

## 3 ROOTS: REPRESENTATION OF OBJECT-ORIENTED 3D SCENES

### 3.1 GENERATIVE PROCESS

The main difference of our model from GQN is that we have a 3D representation *per object* present in the target 3D space while GQN has a single 3D representation compressing the whole 3D space into a vector without object-level decomposition. We begin this modeling by introducing the number of objects $M$ in the target space as a random variable. Then, we can write the representation

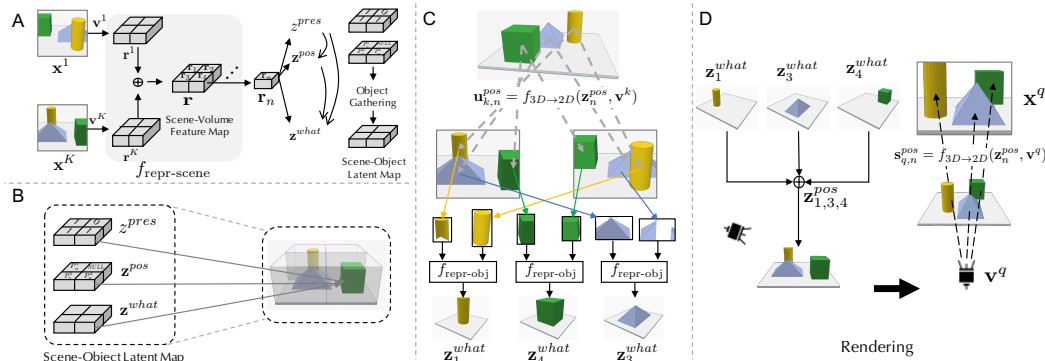

**Figure 1:** Overview of ROOTS. (A) The *context* observations are first fed into an encoder and obtain the scene-volume feature-map. (B) The scene-object latent-map has a cell of $\mathbf{z}_n = z_n^{pres}, \mathbf{z}_n^{pos}, \mathbf{z}_n^{what}$ for each spatial volume cell of the target environment. (C) After obtaining $z_n^{pres}, \mathbf{z}_n^{pos}$, we obtain $\mathbf{z}_n^{what}$ by finding object patches from context and then clustering them for each object. The resulting representation $\mathbf{z}_n^{what}$ is not a 3D-representation. (D) Decoding process $p(\mathbf{x}^q | \mathbf{s}) p(\mathbf{s} | \mathbf{z}, \mathbf{v}^q)$. The representation is rearranged according to query viewpoint $\mathbf{v}^q$ by using $f_{3D \to 2D}(\mathbf{z}_n^{pos}, \mathbf{v}^q)$ and then projected to a target 2D image. In (D), left-bottom is an example of 3D full view. A projection camera is shown on the left corner. On the right, we perform 2 projection steps: (1) converting the coordinate w.r.t. the projection camera (bottom) and (2) projecting onto 2D canvas (up).

prior of ROOTS as $p(\mathbf{z}, M | C) = p(M | C) \prod_{m=1}^{M} p(\mathbf{z}_{(m)} | C)$. To implement such a model with a variable number of objects, in AIR (Eslami et al., 2016), the authors proposed to use an RNN that rolls out $M$ steps, processing one object per step. However, according to our preliminary investigation (under review) and other works (Crawford & Pineau, 2019), it turned out that this approach is computationally inefficient and shows severe performance degradation with growing $M$.

**Object-Factorized Conditional Prior.** To resolve this problem, in ROOTS we propose to process objects in a *spatially local and parallel* way instead of sequential processing. This is done by first introducing the *scene-volume feature-map*. The scene-volume feature-map is obtained by encoding context $C$ into a 3D tensor of $N = (H \times W \times L)$ cells. Each cell $n \in \{1, \dots, N\}$ is then associated to $D$-dimensional *volume feature* $\mathbf{r}_n \in \mathbb{R}^D$. Thus, the actual output of the encoder is a 4-dimensional tensor $\mathbf{r} = f_{\text{repr-scene}}(C)$. Each volume feature $\mathbf{r}_n$ represents a local 3D space in the target 3D space in a similar way that a feature vector in a 2D feature-map of CNN models a local area of an input image. Note, however, that the introduction of the scene-volume feature-map is not the same as the feature-map of 2D images because, unlike the CNN feature-map for images, the actual 3D target space is not directly observable—it is only observed through a proxy of 2D images. For the detail implementation of the encoder $f_{\text{repr-scene}}$, refer to the Appendix A.3.

Given the scene-volume feature-map, for each volume cell $n = 1, \dots, N$ but in parallel, we obtain three latent variables $(z_n^{pres}, \mathbf{z}_n^{pos}, \mathbf{z}_n^{what}) = \mathbf{z}_n$ from the 3D-object prior model $p(\mathbf{z}_n | \mathbf{r}_n)$. We refer this collection of object latent variables $\mathbf{z} = \{\mathbf{z}_n\}_{n=1}^{N}$ to the *scene-object latent-map* as $\mathbf{z}$ is generated from the scene-volume feature-map $\mathbf{r}$. Here, $z_n^{pres}$ is a Bernoulli random variable indicating whether an object is associated (present) to the volume cell or not, $\mathbf{z}_n^{pos}$ is a 3-dimensional coordinate indicating the position of an object in the target 3D space, and $\mathbf{z}_n^{what}$ is a representation vector for the appearance of the object. We defer a more detail description of the 3D-object prior model $p(\mathbf{z}_n | \mathbf{r}_n)$ to the next section. Note that in ROOTS we obtain $\mathbf{z}_n^{what}$ as a 3D representation which is invariant to 3D viewpoints. The position and appearance latents for cell $n$ are defined only when the cell has an associated object to represent, i.e., $z_n^{pres} = 1$. From this modeling using scene-volume feature-map, we can obtain $M = \sum_n z_n^{pres}$ and the previous prior model can be written as follows: $p(\mathbf{z}, M | C) = p(M | C) \prod_{m=1}^{M} p(\mathbf{z}_{(m)} | C) =$

$$\prod_{n=1}^{N} p(\mathbf{z}_n | \mathbf{r}_n) = \prod_{n=1}^{N} p(z_n^{pres} | \mathbf{r}_n) \left[ p(\mathbf{z}_n^{pos} | \mathbf{r}_n) p(\mathbf{z}_n^{what} | \mathbf{r}_n, \mathbf{z}_n^{pos}) \right]^{z_n^{pres}}. \tag{1}$$

In addition to allowing spatially parallel and local object processing, another key idea behind introducing a presence variable per volume cell is to reflect the inductive bias of physics: two objects cannot co-exist at the same position. This helps remove the sequential object processing because dealing with an object does not need to consider other objects if their features are from spatially

distant areas. Note also that the scene-volume feature-map is not to strictly *partition* the target 3D space and that the presence variable represents the existence of the center position of an object not the full volume of an object. Thus, information about an object can exist across neighboring cells.

**Hierarchical Object-Oriented Representation.** The object-oriented representations $\mathbf{z} = \{\mathbf{z}_n\}$ provided by the above prior model is *global* in the sense that it contains all objects in the whole target 3D space, independently to a query viewpoint. From this global representation and given a query viewpoint, ROOTS generates a 2D image corresponding to a query viewpoint. This is done first by learning the view-dependent representation of the target image. In a naive approach, this may be done simply by learning a single vector representation $p(\mathbf{z}^q | \mathbf{z}, \mathbf{v}^q)$ but in this case, we lose important information: the correspondence between a rendered object in the image and a global object representation $\mathbf{z}_n$. That is, we cannot track from which object representation $\mathbf{z}_n$ an object in the image is rendered. In ROOTS, we resolve this problem by introducing *local* 2D-level object-oriented representation layer. This local object-oriented and *view-dependent* representation allows additional useful structure and more interpretability. This 2D local representation is similar to those in AIR (Eslami et al., 2016) and SPAIR (Crawford & Pineau, 2019).

Specifically, for $n$ for which $z_n^{pres} = 1$, a local object representation $\mathbf{s}_n$ is generated by conditioning on the global representation set $\mathbf{z}$ and the query $\mathbf{v}^q$. Our local object representation model is written as: $p(\mathbf{s} | \mathbf{z}, \mathbf{v}^q) = \prod_{n=1}^N p(\mathbf{s}_n | \mathbf{z}, \mathbf{v}^q)$. Similar to the decomposition of $\mathbf{z}_n$, local object representation $\mathbf{s}_n$ consists of $(s_n^{pres}, \mathbf{s}_n^{pos}, \mathbf{s}_n^{scale}, \mathbf{s}_n^{what})$. Here, $s_n^{pres}$ indicates whether an object $n$ should be rendered in the target image from the perspective of the query. Thus, even if an object exists in the target 3D space, i.e., $z_n^{pres} = 1$, $s_n^{pres}$ can be set to zero if that object should be invisible from the query viewpoint. Similarly, $\mathbf{s}_n^{pos}$ and $\mathbf{s}_n^{scale}$ represent respectively the position and scale of object $n$ in the image not in the 3D space, and $\mathbf{s}_n^{what}$ represents the appearance to be rendered into the image (thus not 3D invariant). For more details about how to obtain $(s_n^{pres}, \mathbf{s}_n^{pos}, \mathbf{s}_n^{scale}, \mathbf{s}_n^{what})$ from $\mathbf{z}$ and $\mathbf{v}^q$, we describe in the next section. Given $\mathbf{s} = \{\mathbf{s}_n\}$, we then render to the canvas to obtain the target image $p(\mathbf{x}^q | \mathbf{s})$. Combining all, the generative process of ROOTS is written as follows:

$$p(\mathbf{x}^q | \mathbf{v}^q, C) = \int p(\mathbf{x}^q | \mathbf{s}) \prod_{n=1}^N p(\mathbf{s}_n | \mathbf{z}, \mathbf{v}^q) \prod_{n=1}^N p(\mathbf{z}_n | C) \, \mathrm{d}\mathbf{z} \mathrm{d}\mathbf{s}. \tag{2}$$

See Figure 1 for the overview of the generation process.

## 3.2 IMPLEMENTATION DETAILS

**Global 3D-object prior.** The 3D-object prior $p(\mathbf{z}_n | \mathbf{r}_n)$ generates three latents $(z_n^{pres}, \mathbf{z}_n^{pos}, \mathbf{z}_n^{what})$ as follows. It first obtains the presence latent from $z_n^{pres} \sim \text{Bernoulli}(f_{\text{pres}}(\mathbf{r}_n))$ and the 3-dimension position latent from $\mathbf{z}_n^{pos} \sim \mathcal{N}(f_{pos}^\mu(\mathbf{r}_n), f_{pos}^\sigma(\mathbf{r}_n))$. Using these two latents, we then obtain the appearance latent $\mathbf{z}_n^{what}$. This process is divided into object gathering and object encoding.

For object gathering, for each context image $\mathbf{x}^k$ we attend and crop a patch that corresponds to object $n$. Specifically, we first notice that using the deterministic camera-coordinate projection function $f_{\text{3D}\rightarrow\text{2D}}$ (which we do not learn), we can project from the perspective of a context viewpoint $\mathbf{v}^k$, a 3D position $\mathbf{z}_n^{pos}$ in the global 3D coordinate system into a 2D position $\mathbf{u}_{k,n}^{pos}$ in context image $\mathbf{x}^k$, i.e., $\mathbf{u}_{k,n}^{pos} = f_{\text{3D}\rightarrow\text{2D}}(\mathbf{z}_n^{pos}, \mathbf{v}^k)$. If object $n$ should be invisible from the viewpoint $\mathbf{v}^k$, its projected 2D position $\mathbf{u}_{k,n}^{pos}$ is out of the image $\mathbf{x}^k$ and we do not crop a patch. For more details about the projection function $f_{\text{3D}\rightarrow\text{2D}}$, refer to Wikipedia (2019). We also predict the bounding box scale $\mathbf{u}_{k,n}^{scale} = f_{\text{scale}}(\mathbf{u}_{k,n}^{pos}, \mathbf{r}_n, \mathbf{v}^k)$. Given the center position and scale, we can crop a patch $\mathbf{x}_n^k \subset \mathbf{x}^k$ using the spatial transformer (Jaderberg et al., 2015). Applying this cropping to all context pairs $(\mathbf{v}^k, \mathbf{x}^k) \in C$, we gather a set of object image-patches $X_n = \{\mathbf{x}_n^k\}_{k=1}^K$ for all $n$ with $z_n^{pres} = 1$.

Given object image-patches $X_n$, obtaining object 3D-representation $\mathbf{z}_n^{what}$ can be converted to a GQN encoding problem. That is, we can simply consider $X_n$ and its corresponding viewpoints as a new object-level context $C_n$, and can run an order-invariant GQN representation network $\mathbf{r}_n^{what} = f_{\text{repr-obj}}(C_n)$ and then run ConvDRAW to obtain $\mathbf{z}_n^{what}$ from $\mathbf{r}_n^{what}$.

**Local 2D-object prior.** The intermediate prior $p(\mathbf{s}_n | \mathbf{z}, \mathbf{v}^q)$ generates $(s_n^{pres}, \mathbf{s}_n^{pos}, \mathbf{s}_n^{scale}, \mathbf{s}_n^{what})$. This is done as follows. Similar to what we described in the above to obtain $\mathbf{u}_{k,n}^{pos}$, we use the coordinate projection function $f_{\text{3D}\rightarrow\text{2D}}$ to find the position of a global object $\mathbf{z}_n$ in the

target image from the perspective of query $\mathbf{v}^q$, i.e., $\mathbf{s}^{pos}_{q,n} = f_{\text{3D}\rightarrow\text{2D}}(\mathbf{z}^{pos}_n, \mathbf{v}^q)$. Thus, we model the position as a deterministic variable. The scale $\mathbf{s}^{scale}_{q,n}$ is also obtained similarly as described in the above for $\mathbf{u}^{scale}_{k,n}$, but with random sampling. Then, we predict $s^{pres}_n \sim$ Bern$(f_{\text{pres}}(\mathbf{s}^{pos}_{q,n}, \mathbf{s}^{scale}_{q,n}, \mathbf{z}_n, \mathbf{v}^q))$. If object $n$ should be visible in the 2D target image, i.e., $s^{pres}_n = 1$, we generate the 2D appearance representation $\mathbf{s}^{what}_n$ by using an object-level GQN decoder based on ConvDRAW, i.e., $\mathbf{s}^{what}_n = \text{ConvDRAW}(\mathbf{z}^{what}_n, \mathbf{v}^q)$. Finally, we have $p(\mathbf{s}_n|\mathbf{z},\mathbf{v}^q) = p(\mathbf{s}^{pos}_n|\mathbf{z}^{pos}_n,\mathbf{v}^q)p(\mathbf{s}^{scale}_n|\mathbf{s}^{pos}_n,\mathbf{z}^{what}_n,\mathbf{v}^q)p(s^{pres}_n|\mathbf{s}^{pos}_n,\mathbf{s}^{scale}_n,\mathbf{z}_n,\mathbf{v}^q)$.

**Rendering to 2D Canvas.** A main challenge in rendering the local representation $\mathbf{s} = \{\mathbf{s}_n\}$ into the image canvas, i.e., $p(\mathbf{x}^q|\mathbf{s})$, is to deal with occlusion. In ROOTS, this can easily be achieved by noticing (i) that the coordinate conversion $f_{\text{3D}\rightarrow\text{2D}}(\mathbf{z}^{pos}_n, \mathbf{v}^q)$ actually converts a 3D-coordinate to another 3D-coordinate and (ii) that then the last dimension of the converted coordinate system can be interpreted as the orthogonal projection distance from the viewpoint $\mathbf{v}^q$ to the object's position $\mathbf{z}^{pos}_n$. We can use this distance as object depth from the observer's perspective. This allows us to sort objects according to their depths and render each object accordingly without occlusion. To handle background, ROOTS has an independent module to infer the background separately at image-level. We also found that learning an object-level mask along with the appearance latent $\mathbf{s}^{what}_n$ helps segmentation between foreground and background as well as generating less blurry images. More details on related implementation are provided in Appendix A.4.

### 3.3 Learning and Inference

Due to the intractability of the posterior $p(\mathbf{z},\mathbf{s}|C,D)$ with $D = \{(\mathbf{v}^q_j, \mathbf{x}^q_j)\}_j$ the target viewpoint-image pairs, we train ROOTS using variational inference with the following posterior approximation $q_\phi(\mathbf{z},\mathbf{s}|C,D) = q_\phi(\mathbf{z}|C,D)q_\phi(\mathbf{s}|\mathbf{z},C,D)$. To compute the gradient w.r.t. the continuous latent variables such as the position and appearance, we use reparameterization trick and for the discrete variables on the presence, we use a continuous relaxation using Gumbel-Softmax trick (Jang et al., 2016). Other methods based on the REINFORCE algorithm (Williams, 1992; Tucker et al., 2017; Grathwohl et al., 2017) can also be used. Implementation of the approximate posterior $q(\mathbf{z}|C,D)$ and $q(\mathbf{s}|\mathbf{z},C,D)$ is made easier by sharing the parameters of $q$ with that of conditional prior $p(\mathbf{z}|C)$: we only need to provide additional data $D$. With $D = (\mathbf{v}^q,\mathbf{x}^q)$ for simplicity, the objective is to maximize the following evidence lower bound (ELBO): $\mathcal{L}(\theta,\phi;C,D) =$

$$\mathbb{E}_{\mathbf{s},\mathbf{z}\sim q_\phi}\left[\log p_\theta(\mathbf{x}^q|\mathbf{s}) - \mathbb{KL}[q_\phi(\mathbf{s}|\mathbf{z},\mathbf{v}^q,C,D) \parallel p_\theta(\mathbf{s}|\mathbf{z},\mathbf{v}^q)]\right] - \mathbb{KL}[q_\phi(\mathbf{z}|C,D) \parallel p_\theta(\mathbf{z}|C)]. \quad (3)$$

**Combining with Unconditioned Prior.** One difficulty in using the conditional prior is the fact that, because the prior is learned, we have less control in reflecting our prior knowledge into the prior distribution. In our experiments, it turns out that biasing the posteriors of some variables towards the values of our prior preference is helpful in stabilizing the model. To implement this, in our training, we use the following objective that has additional KL terms between the posterior and *unconditioned* prior.

$$\begin{aligned}
\mathcal{L} = \mathcal{L}(\theta,\phi;C,D) &+ \mathbb{KL}[q_\phi(\mathbf{z}^{pos},\mathbf{s}^{scale}|C,D) \parallel \mathcal{N}(0,\mathbb{I})] \\
&+ \gamma\mathbb{KL}[q_\phi(\mathbf{z}^{pres}|C,D) \parallel \text{Geom}(\rho)] + \gamma\mathbb{E}_{\mathbf{z}\sim q_\phi}[\mathbb{KL}[q_\phi(\mathbf{s}^{pres}|\mathbf{z}^{pres},C,D) \parallel \text{Geom}(\rho)]]
\end{aligned} \quad (4)$$

where $\mathbf{z}^{pres} = \{z^{pres}_n\}_n$ and $\mathbf{z}^{pos},\mathbf{z}^{scale},\mathbf{s}^{pres}$ are defined similarly. The $\rho$ and $\gamma$ are hyperparameters weighting the auxiliary loss terms. We set them to 0.999 and 7 during training, respectively. This auxiliary loss can be derived if we replace the conditional prior by the product of expert prior $p(\mathbf{z}|C)p(\mathbf{z})$ divided by the posterior $q(\mathbf{z}|C)$.

## 4 Related Works

Although to our knowledge there has been no previous work on unsupervised and probabilistic generative object-oriented representation learning for 3D scenes containing multiple objects, there has been literature on its 2D problems. The first is the Attend, Infer, Repeat (AIR) model (Eslami et al., 2016). AIR uses spatial transformer (Jaderberg et al., 2015) to crop an object patch and uses an RNN to sequentially generate next object conditioning on the previous objects. In Crawford & Pineau (2019), the authors showed that this RNN-based rollout is inefficient and can significantly

degrade performance as the number of objects increases. SPAIR is inspired by YOLO (Redmon et al., 2016), but unlike YOLO, it does not require bounding box labels. The main limitation of SPAIR is to infer the latents sequentially. Neural Expectation Maximization (NEM) (Greff et al., 2017) considers the observed image as a pixel-level mixture of $K$ objects and an image per object is generated and combined according to the mixture probability. In Greff et al. (2019), the authors proposed a more efficient version of NEM, called IODINE, using iterative inference (Marino et al., 2018). In MONET (Burgess et al., 2019), an RNN drawing a scene component at each time step is used. Unlike AIR and SPAIR, the representations in NEM, IODINE, and MONET do not explicitly provide natural disentanglement like presence and pose per object.

There have been plenty amount of remarkable works about visual 3D learning from the computer vision community. However, as pointed in Section 1, the problem setting and proposed model of these works are different from ours in the sense that they are either (i) not decomposing object-wise representations from a scene containing multiple objects (but working mostly on single object cases) (Wu et al., 2016; Yan et al., 2016; Choy et al., 2016; Kar et al., 2017; Nguyen-Phuoc et al., 2019), (ii) supervised approach (Huang et al., 2018; Tulsiani et al., 2018; Cheng et al., 2018; Shin et al., 2019; Du et al., 2018), (iii) learn image generation (synthesis) of a 3D scene without scene-representation (Sitzmann et al., 2019; Kato & Harada, 2019; Pinheiro et al., 2019; Tulsiani et al., 2017) or learn scene-representation without learning rendering ability (Zhou et al., 2017; Yu & Wang, 2018), or (iv) not end-to-end. Many of the more traditional works from 3D computer vision also relevant to our work but many of them are not based on neural networks and not end-to-end trainable.

## 5 EXPERIMENTS

We evaluate ROOTS quantitatively and qualitatively. We train ROOTS with the same hyperparameters on all datasets. We first briefly describe the datasets. For more details of the dataset generation and network architecture, refer to Appendix A.5 and Appendix A.4. We use MuJoCo, (Todorov et al., 2012), to simulate 3D scenes. Specifically, we generate three different datasets of scenes with 1-3 objects, 2-4 objects and 3-5 objects, respectively. We set the image size to $64 \times 64 \times 3$ pixels. For each object, we randomly choose its position, shape, size, and color. Although we put all objects on the floor, we still predict the 3-dimension coordinate values for $\mathbf{z}_n^{pos}$ because objects have different sizes. We generate 60K different scenes for each dataset and split them into 50k for training, 5k for validation, and 5k for testing. We use CGQN as the baseline implementation but in the rest of the section use the abbreviation 'GQN' instead of 'CGQN' to indicate the general GQN framework.

### 5.1 QUALITATIVE EVALUATION

First, we compare the generations between ROOTS with GQN by visualizing several generated samples from the same scene under the same set of query viewpoints. Note that the goal of this experiment is to show that achieving the object-wise representation in ROOTS does not deteriorate its generation quality in comparison to GQN. As seen in Figure 2, ROOTS generates slightly sharper object boundaries while GQN generates more blurry images. We believe that this is due to the object-wise generation using segmentation masks. Then, to provide a further understanding of the advantages of object-orientated representation, we visualize decomposed generations from ROOTS, as shown in Figure 3.A. We can see that ROOTS can generate clean background, complete objects, precise foreground, and detailed occlusion mask separately. Furthermore, to demonstrate the viewpoint-invariant property of the global object representation $\mathbf{z}_n^{what}$, in Figure 3.B, we provide generations of two objects under different query viewpoints. We can see that ROOTS recovers the object with pose corresponding to query viewpoints. GQN cannot provide such decomposed generations because its representation is scene-level where objects are not decomposed.

**Object-wise disentanglement.** *Generations from random viewpoints after changing object positions.* In this section and the following sections, we use ROOTS trained on the 2-4 object dataset for demonstration unless otherwise stated. To verify the disentanglement property of our object-oriented representation learned by ROOTS, we carry out the experiment of arbitrarily modifying $\mathbf{z}_n^{pos}$ of one object in a scene. As a good disentangled latent representation, this modification should not affect the position, existence, and appearance of other objects concurrent in the scene while handling occlusion properly. More importantly, we should be able to change either the $x$ or $y$ coordinate independently and this change should be consistent across viewpoints. To demonstrate this,

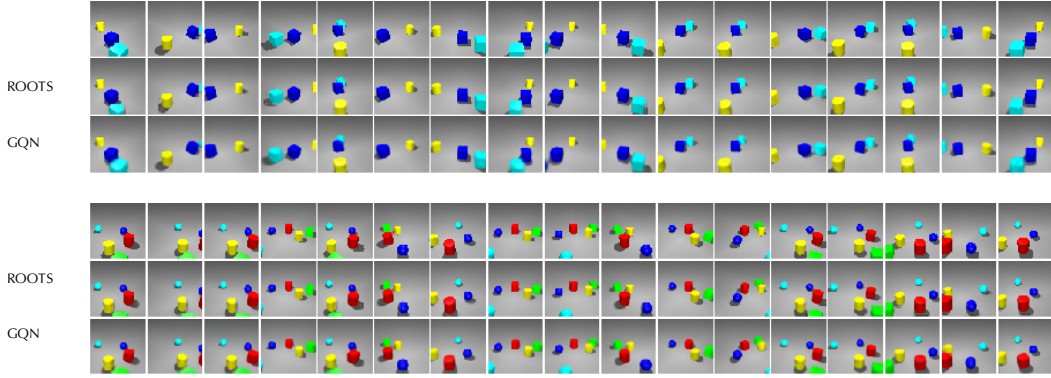

**Figure 2:** Examples of generations from two different scenes together with ground truth placed in the first row for each scene. ROOTS shows better generation on objects boundary and clearer occlusion, compared with GQN, especially when there are more object in the scene.

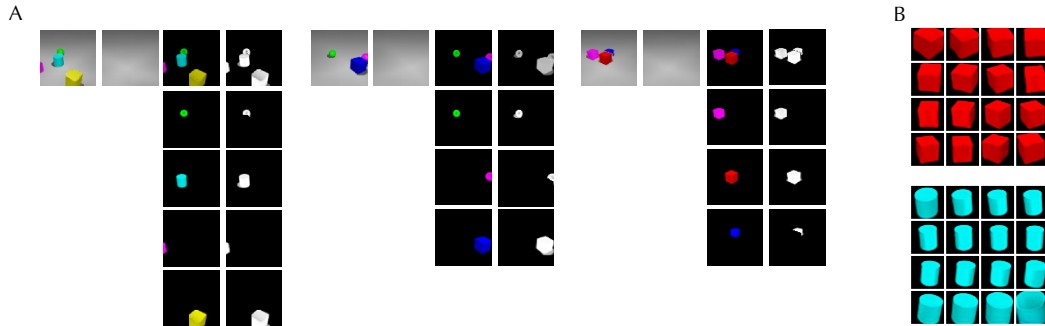

**Figure 3: A**: Generated examples for scene decomposition from three different scenes visualized as 2D images captured by a query camera (viewpoint). For example, in the first group of visualization in A, ROOTS can segment a scene into the foreground and background first and decompose foreground into each individual object further, which are a green sphere, a blue cylinder, a partially observed cube and a yellow cube. We also show the foreground occlusion mask, which is obtained via each object's mask and the distance between the object and query camera. **B**: Recovered generations of two objects from global 3D object-wise representation under different query viewpoints. The recovered 2D projection have different pose when seen from different viewpoints.

after changing one dimension of $\mathbf{z}_n^{pos}$ of the red cylinder, as shown in Figure 4, we feed this modified $\hat{\mathbf{z}}_n^{pos}$ back to the model and follow the ROOTS generation model in the same way as we do during testing. Generations under 4 different viewpoints are shown in Figure 4. We can see that ROOTS has a strong ability to learn disentangled representation and occlusions have been handled well due to this advantage.

**Compositionality.** As stated in earlier sections that a 3D scene can be decomposed into several independent objects, another advantage coming with object-orientated representations is that a new scene can be easily built up with selected object components. Thus, by simple combination, we can build novel scenes. To demonstrate this, we first provide ROOTS with three sets of context images from three different scenes and save the learned object representations $\mathbf{z}$ for each scene, $\mathbf{z} = \{\mathbf{z}_n^{pos}, \mathbf{z}_n^{what}\}$ for all $n$ with $\mathbf{z}_n^{pos} = 1$. Then we swap one object between the first two scenes and add one additional object to the third scene, as shown in Figure 5. We see that the object component learned by ROOTS is fully disentangled and can be reused to create new scenes. Also, by adding one new object, we make a scene with 5 objects, which does not exist in the training dataset. Another example shown a scene 9 objects is visualized in Figure 6, where we use ROOTS trained on the 1-3 objects dataset. More details about this experiment can be found in Appendix A.2.

**Partial Observability.** We know that the successful generation of ROOTS, even with only partial observations provided, is coming from object gathering across viewpoints. This is because if an object is invisible for one viewpoint, ROOTS can learn it from other viewpoints. Here, we want

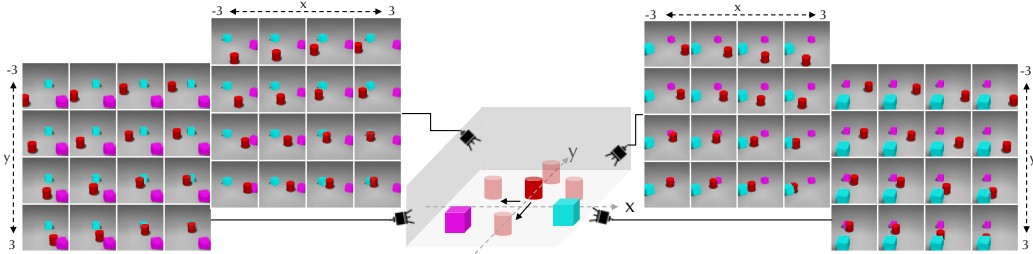

**Figure 4:** Visualization of changing $\mathbf{z}_n^{pos,x}$ and $\mathbf{z}_n^{pos,y}$ of the red cylinder in a scene through generations from 4 different cameras. We also simulated the 3D scene in the center. Cameras are shown as an example. We put walls for readers to better understand the relative height of cameras. There is no wall in the real dataset.

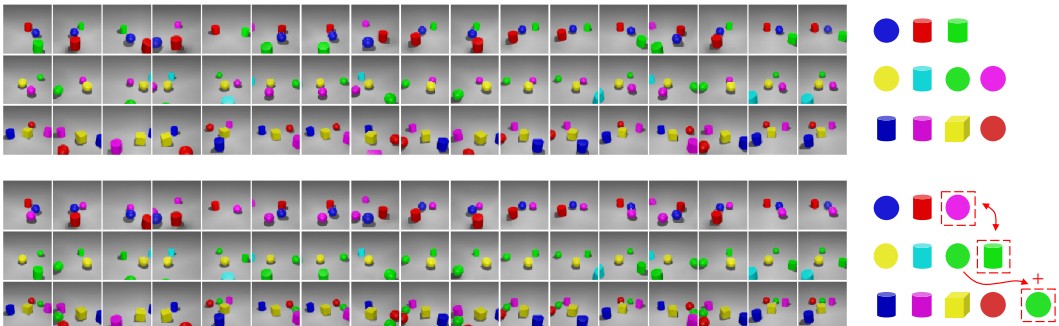

**Figure 5:** TOP: Original generations from three different scenes, icons on the right side highlight objects in each scene. Each column shows a 2D image captured by one camera. Bottom: Manipulated generations from the same three scenes under the same set of cameras. We switch the green cylinder from scene 1 with the purple sphere from scene 2 and we copy green sphere from scene 2 and put it into scene 3.

to push the partial observability to the extreme case, where one object is totally invisible for all context viewpoints. In this case, ROOTS should not be able to correctly predict its existence, just like humans observing the true physical world. To show this, we manually select some images from a scene that has one object missing served as context and the rest served as targets for ROOTS to generate. We show the results in Figure 7.

## 5.2 QUANTITATIVE EVALUATION

**NLL and MSE.** In this section, we compare the quantitative results of ROOTS and GQN on the negative log-likelihood (NLL) and the mean squared error (MSE) on the test dataset. The goal is to show quantitatively that achieving the object-wise representation in ROOTS does not deteriorate its generation quality in comparison to GQN. We approximate NLL using importance sampling with $K = 50$ samples and report the image NLL normalized by the number of pixels in Table **??**. Both GQN and ROOTS are trained for 120 epochs on each dataset, making sure both ROOTS and GQN have converged. We see that, although learning object-factorized representations, ROOTS achieves NLL comparable to GQN. Due to object-wise representation, ROOTS recovers objects separately, together with an independent background module, ROOTS performs better on MSE than GQN. This benefit becomes clearer as the number of objects in the scene grows.

**Table 1:** Negative log-likelihood and mean squared error

| Training set | 1-3 objects | | 2-4 objects | | 3-5 objects | |
|---|---|---|---|---|---|---|
| Metrics | NLL | MSE | NLL | MSE | NLL | MSE |
| ROOTS | 0.9199 | 15.16 | 0.9205 | 25.37 | 0.9213 | 29.82 |
| GQN | 0.9196 | 15.28 | 0.9201 | 26.86 | 0.9204 | 32.66 |

**Object Detection Quality.** In this section, we only provide precision and recall results as object detection evaluation of ROOTS as GQN cannot detect objects. To estimate the true positive prediction, we first filter out predictions that are too far away from any ground truth by applying a

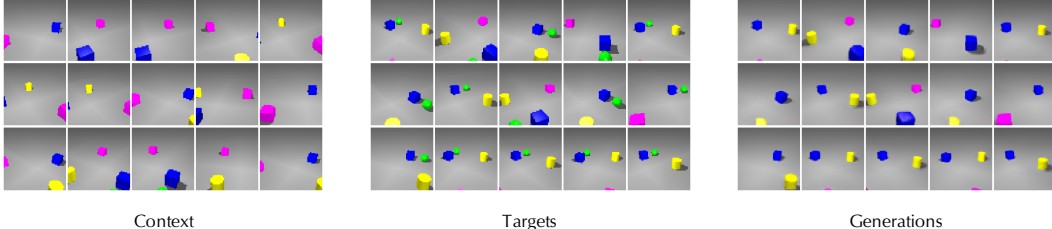

**Figure 6:** A novel scene consisted of 9 objects is composited by ROOTS trained on 1-3 objects dataset.

Context | Targets | Generations

**Figure 7:** Left: Context images provided to ROOTS, we select images that do not have the green sphere projected. Middle: Target images, from which we can see that there is one green sphere existed. Right: Generations from ROOTS given context images from the left.

radius threshold. The distance is calculated as the euclidean distance between two center points. Then we assign the predicted object to the ground truth object to which it has the nearest distance. Multi-association is not allowed. We normalize the coordinate value to be in the range of $[-1, 1]$ for better interpretability of the applied radius threshold. Accordingly, the average object size is $0.3$. We provide the precision and recall result under different thresholds in Table **??**. Besides precision and recall, we also provide the counting accuracy of ROOTS.

**Table 2:** Precision and Recall

| Training set | 1-3 objects | | | 2-4 objects | | | 3-5 objects | | |
|---|---|---|---|---|---|---|---|---|---|
| Threshold | 0.1 | 0.15 | 0.35 | 0.1 | 0.15 | 0.35 | 0.1 | 0.15 | 0.35 |
| Precision | 76.82 | 91.56 | 98.68 | 66.76 | 86.07 | 98.29 | 66.64 | 86.09 | 96.65 |
| Recall | 75.56 | 90.15 | 97.40 | 65.53 | 84.57 | 95.93 | 66.91 | 86.47 | 97.10 |
| Count Acc. | | 93.18 | | | 88.16 | | | 82.21 | |

## 6 CONCLUSION

We proposed ROOTS, a probabilistic generative model for unsupervised learning of 3D scene representation and rendering. ROOTS can learn object-oriented interpretable and hierarchical 3D scene representation. In experiments, we showed the generation, decomposition, and detection ability of ROOTS. For compositionality and transferability, we also showed that, due to the factorization of structured representation, new scenes can be easily built up by reusing components from 3D scenes. Interesting future directions would be to learn the knowledge of the 3D world in a sequential manner as we humans keep updating and inferring knowledge through time.

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

# A   APPENDIX

## A.1   GENERALIZATION

In this section, we quantitatively analyze ROOTS's generalization performance on tasks of generation and detection. For the task of generation, we provide a comparison between ROOTS and GQN. Specifically, we train both ROOTS and GQN on our three datasets until full convergence. For each model trained on one dataset, e.g 1-3 objects dataset, we test it on the other two datasets, e.g 2-4 objects dataset, and 3-5 objects dataset. For the task of detection, since GQN only provide scene-level representation (detection is not possible), we only report detection performance of ROOTS. Generation results (MSE) and detection results (precisin and recall) are shown in Table 3 and Table 4, respectively. Precision and recall are reported with the radius threshold set to 0.15 unit in Mujoco physic world. Here, we describe the trend we find from the experimental observations. **When trained on dataset with small (e.g., 1-3) number of objects and tested on large (e.g., 3-5) number of objects**, both ROOTS and GQN do not generalize well. This is because, during testing, latent variables are sampled from a learned prior. If during training, the model isn't provided a chance to see scenes with more objects, it is hard to generalize well. Also, we note that, compared with GQN, ROOTS obtains larger MSE error. Together with observations in the following case, we hypothesize this sensitivity comes from the discrete variable, presenting the existence of an object. While with one vector representing the full scene, GQN has smoother performance. **When trained on dataset with a large number of objects while tested on a small number of objects**, ROOTS and GQN show a similar property in their performance. They both perform well. But, in this case, ROOTS produces better generations results, which we believe is a benefit from the object-wise generation and a separate background module. While in GQN all information are compressed into one vector. The above conclusion of ROOTS is consistent with its performance on detection task. One interesting finding during this experiment is that ROOTS trained on 3-5 objects dataset achieves better results (lower MSE, higher precision, and recall) than trained on 1-3 objects dataset when testing on 1-3 objects dataset. A similar phenomenon is found when testing on 2-4 objects dataset. This is an interesting finding that is worth more investigation.

**Table 3:** Quantitative generalization results on generation

| Training set | 1-3 Objects | | | 2-4 Objects | | | 3-5 Objects | | |
|---|---|---|---|---|---|---|---|---|---|
| Testing set | 1-3 Objects | 2-4 Objects | 3-5 Objects | 1-3 Objects | 2-4 Objects | 3-5 Objects | 1-3 Objects | 2-4 Objects | 3-5 Objects |
| GQN | 15.57 | 27.49 | 43.57 | 16.44 | 26.77 | 37.62 | 15.02 | 23.73 | 33.38 |
| ROOTS | 14.72 | 35.22 | 71.43 | 13.97 | 25.35 | 42.21 | 11.48 | 19.62 | 29.88 |

**Table 4:** Quantitative generalization results on object detection

| Training set | 1-3 Objects | | | 2-4 Objects | | | 3-5 Objects | | |
|---|---|---|---|---|---|---|---|---|---|
| Testing set | 1-3 Objects | 2-4 Objects | 3-5 Objects | 1-3 Objects | 2-4 Objects | 3-5 Objects | 1-3 Objects | 2-4 Objects | 3-5 Objects |
| Precision | 91.43 | 83.25 | 71.81 | 90.56 | 86.55 | 80.83 | 93.24 | 90.10 | 85.91 |
| Recall | 89.84 | 80.02 | 65.67 | 89.22 | 84.68 | 78.29 | 93.66 | 89.94 | 85.91 |
| Count Acc. | 93.07 | 81.04 | 57.91 | 93.17 | 87.42 | 79.42 | 92.57 | 88.14 | 82.17 |

## A.2   COMPOSING WITH MORE OBJECTS

In this section, we provide one more example of compositionality. We first train ROOTS on 1-3 objects dataset. As in Figure 8, during testing, we collect object representations from 7 different scenes and reuse them to composite a new scene with 9 objects. Additionally, the sampled center position of each object and query viewpoints (represented as camera position, pitch, and roll) fed into ROOTS are with respect to canonical coordinates. Note that, to predict the scale of objects in 2D projection correctly (e.g., for the same object, the larger the distance between it and viewpoint is, the smaller that object should be in the 2D projection), ROOTS learns to infer local depth (position translation) for each object given specific query viewpoint. This can be observed on the yellow sphere, green cylinder and blue cube in the bottom row in Figure 8.

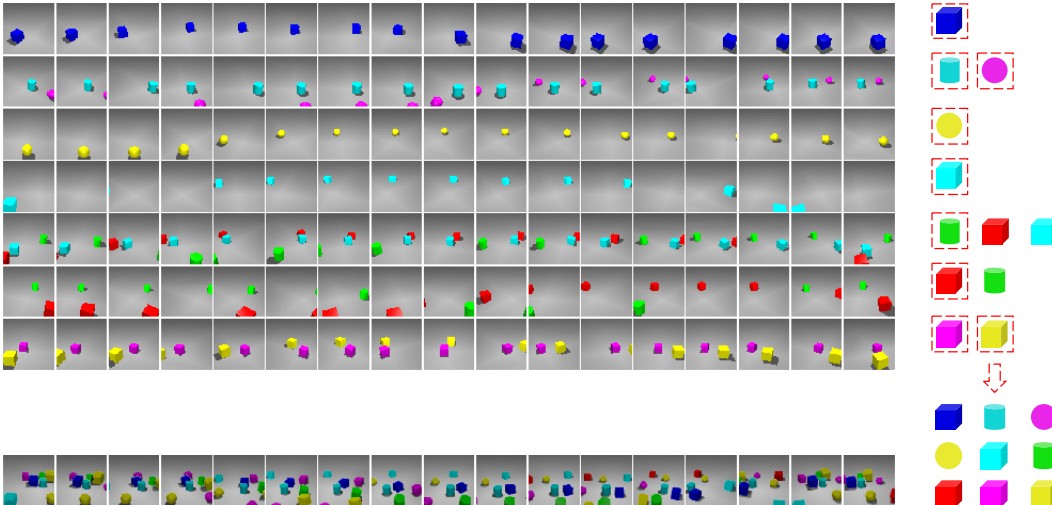

**Figure 8:** TOP: Target images taken by several query viewpoints from 9 different scenes with icons on the right hand-side highlighting the objects in the scene. Bottom: New scene created with objects collected from the above 9 scenes.

### A.3 DETAILS OF COMPONENT MODULES

We first introduce some important building-blocks for implementing our model in this section, and then sketch the implementation steps using modules described here in the following section.

**Scene Representation Network:** We use Scene Representation Network to implement the order invariant encoder at scene level, $f_{\text{repr-scene}}(\cdot)$. This network is modified based on Representation Network in GQN. We adjust kernel size and add a CNN3D layer to make sure the output conv-features fit our needs, shown as in Figure 9. The scene representation network takes <image, viewpoint> pair as input, output a conv-feature map. To implement the order invariant, we take the mean of these outputs from the same scene.

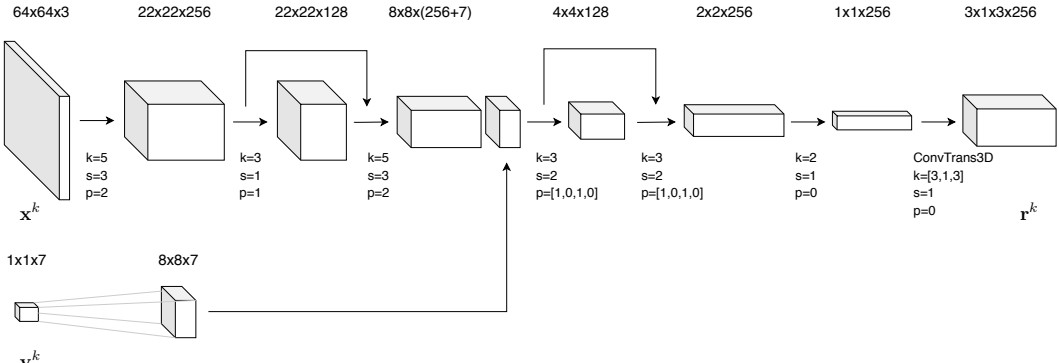

**Figure 9:** Scene Representation Network

**Object Representation Network:** We use object Representation Network to implement order invariant encoder at object level, $f_{\text{repr-obj}}(\cdot)$. As visualized in Figure 10, we design a branch to pass low level feature to provide richer conv-features. The dimension $d$ in the second input equals to the summation of the dimension of $\{z^{pres}, s^{scale}, \text{and } s^{pos}\}$. The corresponding values are listed in Table 5. The order invariant is implemented in the same way as in Scene Representation Network.

**Convolutional LSTM Cell:** We use ConvLSTM as the RNN module in the ConvDRAW module. In ConvLSTM, all fully-connected layers are substituted with convolutional layers. Its one updating step is described as follows:

$$(h_{i+1}, c_{i+1}) \longleftarrow \text{ConvLSTM}(\mathbf{x}_i, h_i, c_i)$$

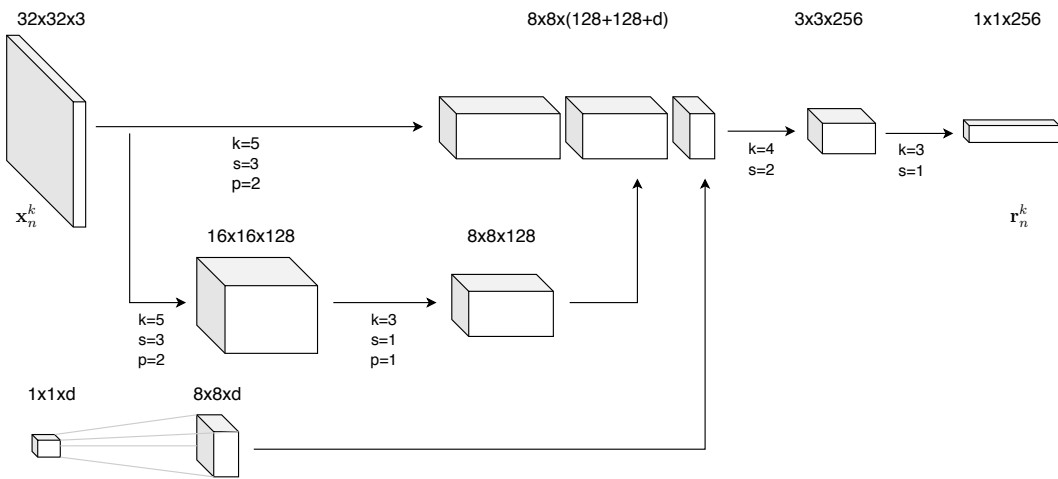

**Figure 10:** Object Representation Network

where $h_i$ is the output of the cell and $c_i$ is the recurrent state of the ConvLSTM, $\mathbf{x}_i$ is the input. Both $h_i$ and $c_i$ are initialized with zeros at the $i = 0$ step.

**ConvDRAW:** We highlight one step ConvDRAW (denoted as $l$) used in ROOTS here for generative process and inference process separately.

- Generative Process:

$$(\mathbf{h}_p^{(l+1)}, \mathbf{c}_p^{(l+1)}) \leftarrow \text{ConvLSTM}_\theta(\mathbf{x}^{(l)}, \mathbf{z}^{(l)}, \mathbf{h}_p^{(l)}, \mathbf{c}_p^{(l)})$$
$$\mathbf{z}^{(l+1)} \sim \text{StatisNet}(\mathbf{h}_p^{(l+1)})$$

- Inference Process:

$$(\mathbf{h}_q^{(l+1)}, \mathbf{c}_q^{(l+1)}) \leftarrow \text{ConvLSTM}_\phi(\mathbf{y}, \mathbf{x}^{(l)}, \mathbf{h}_p^{(l)}, \mathbf{h}_q^{(l)}, \mathbf{c}_q^{(l)})$$
$$\mathbf{z}^{(l+1)} \sim \text{StatisNet}(\mathbf{h}_q^{(l+1)})$$
$$(\mathbf{h}_p^{(l+1)}, \mathbf{c}_p^{(l+1)}) \leftarrow \text{ConvLSTM}_\theta(\mathbf{e}^{(l)}, \mathbf{z}^{(l)}, \mathbf{h}_p^{(l)}, \mathbf{c}_p^{(l)})$$

where $\mathbf{x}^l$ is the input at the $l_{th}$ step, $\mathbf{y}$ is target reconstruction, $z^{(l+1)}$ is the sampled latent at the $l$th step. We denote the prior module and posterior module with subscript $p$ and $q$, respectively. $\theta$ and $\phi$ are neural network parameters. The StatisNet is described in the following paragraph. In the Renderer network, we replace the StatisNet with a deterministic Decoder.

**Sufficient Statistic Networks:** The Sufficient Statistic Networks will output sufficient statistics for pre-defined distribution, e.g. $\mu$ and $\sigma$ for Gaussian distribution, given inputs. We list the configuration of all the Sufficient Statistics Networks in Table 5 used during generation. For $z_n^{pos}$, $z_n^{what}$ and $s_{n,i}^{scale}$, we use auto-regressive scheme (ConvDraw) to learn the sufficient statistics. For $z_n^{pres}$ and $s_{q,n}^{pres}$, we use regular convolutional layers. In the third column, we give the kernel size of the first convolutional layer, the remaining are Conv3D 1X1 layers. If the kernel size is 3, we have one zero paddings, the stride is 1 to keep the spatial size unchanged. The "Draw Rollouts" column shows the number of DRAW steps we apply. The "Concat" column specifies how we use sampled latent value for the subsequent process, for example, concatenating $z^l$, for $l < L$ or only taking $z^L$, where $L$ is the rollout steps.

**GQN Implementation:** We strictly follow the paper for implementation details. The only difference is we enhance the decoder in the renderer by adding two more convolutional layers.

A.4 IMPLEMENTATION DETAILS

Here, we give the details of the implementation of ROOTS. Details of component modules are mentioned in Appendix A.3. We first outline the generation and inference process for one object, indexed with $n$. Parallelizing it to multiple objects is straightforward.

**Table 5:** Configuration of Sufficient Statistic Networks

| Latent Variable | Channel Numbers | K | Draw Rollouts | Concate |
|---|---|---|---|---|
| $\mathbf{z}_n^{pos}$ | [128, 128, 64, 32, 3] | 3 | 2 | last step |
| $\mathbf{z}_n^{what}$ | [128, 128, 64, 32, 4] | 1 | 4 | concatenate |
| $\mathbf{s}_{q,n}^{scale}$ | [128, 128, 64, 32, 2] | 1 | 4 | last step |
| $z_n^{pres}$ | [256, 256, 128, 64, 1] | 3 | - | - |
| $s_{q,n}^{pres}$ | [271, 256, 128, 64, 32, 1] | 1 | - | - |
| $\mathbf{z}^{bg}$ | [128, 4] | 3 | 2 | last step |

**Generation Process** Superscript on ConvDRAW is used to indicate which variable the ConvDRAW module is responsible for. The Renderer module is implemented in the same way as ConvDRAW with one difference that we do not model any variable in Renderer, making it a deterministic decoder. All the ConvDRAW modules have a hidden state size of 128. We use $\mathcal{ST}$ denote spatial transformation process and $\mathcal{ST}^{-1}$ denotes the reversed process. $K$ denote the index set of context C and $\theta$ and $\phi$ are neural network parameters.

$$\mathbf{r} \leftarrow \sum_{k \in K} f_{\text{repr-scene}}(\mathbf{x}^k, \mathbf{v}^k, \theta) \qquad \text{(Obtain scene-volume feature-map)} \qquad (5)$$

$$\mathbf{z}_n^{pos} \sim \text{ConvDRAW}_\theta^{pos}(\mathbf{r}_n) \qquad \text{(Sample 3D position)} \qquad (6)$$

$$z_n^{pres} \sim \text{CONV}_\theta(\mathbf{r}_n) \qquad \text{(Sample global presence)} \qquad (7)$$

$$\mathbf{u}_{k,n}^{pos} \leftarrow f_{3D \rightarrow 2D}(\mathbf{z}_n^{pos}, \mathbf{v}^k), k \in K \qquad \text{(Perspective projection)} \qquad (8)$$

$$\mathbf{u}_{k,n}^{scale} \sim \text{ConvDRAW}_\theta^{scale}(\mathbf{r}_n, \mathbf{v}^k, \mathbf{u}_{k,n}^{pos}), k \in K \qquad \text{(Sample object scale)} \qquad (9)$$

$$\mathbf{x}_n^k \leftarrow \mathcal{ST}(\mathbf{x}^k, [\mathbf{u}_{k,n}^{pos}, \mathbf{u}_{k,n}^{scale}]), k \in K \qquad \text{(Crop object patch)} \qquad (10)$$

$$\mathbf{r}_n^{what} \leftarrow \sum_{k \in K} f_{\text{repr-obj}}(\mathbf{x}_n^k, \mathbf{v}_k), k \in K \qquad \text{(Object level encoding)} \qquad (11)$$

$$\mathbf{z}_n^{what} \sim \text{ConvDRAW}_\theta^{what}(\mathbf{r}_n^{what}) \qquad \text{(Sample global what)} \qquad (12)$$

$$\mathbf{s}_{q,n}^{pos} \leftarrow f_{3D \rightarrow 2D}(z_n^{pos}, \mathbf{v}^q) \qquad \text{(Perspective projection)} \qquad (13)$$

$$\mathbf{s}_{q,n}^{scale} \sim \text{ConvDRAW}_\theta^{scale}(\mathbf{r}_n, \mathbf{s}_{q,n}^{pos}, \mathbf{v}^q) \qquad \text{(Sample object scale)} \qquad (14)$$

$$s_{q,n}^{pres} \sim \text{CONV}_\theta(z_n^{pres}, \mathbf{r}_n^{what}, \mathbf{v}^q, \mathbf{s}_{q,n}^{pos}, \mathbf{s}_{q,n}^{scale}) \qquad \text{(Sample local presence)} \qquad (15)$$

$$\hat{\mathbf{x}}_n^q, \alpha_n^q \leftarrow \text{Renderer}_\theta(\mathbf{z}_n^{what}, \mathbf{v}^q) \qquad \text{(Decode the object patch and object mask)} \qquad (16)$$

$$\hat{\mathbf{x}}_n^q \leftarrow \mathcal{ST}^{-1}(\alpha_n^q \times \hat{\mathbf{x}}_n^q, [\mathbf{s}_{q,n}^{pos}, \mathbf{s}_{q,n}^{scale}]) \qquad \text{(Generate object patch)} \qquad (17)$$

$$\alpha_n^q \leftarrow \mathcal{ST}^{-1}(\alpha_n^q, [\mathbf{s}_n^{q\,pos}, \mathbf{s}_n^{q\,scale}]) \qquad (18)$$

$$\alpha_{q,n}^{occ} \leftarrow depth_n \times \alpha_n^q == \min_n(depth_n \times \alpha_n^q) \qquad \text{(Obtain occlusion mask)} \qquad (19)$$

$$\hat{\mathbf{x}}_{fg}^q \leftarrow \sum_n (\alpha_{q,n}^{occ} \times \hat{\mathbf{x}}_n^q \times s_{q,n}^{pres}) \qquad \text{(Generate foreground)} \qquad (20)$$

$$\alpha_{fg}^q \leftarrow \sum_n (\alpha_n^q \times \alpha_{q,n}^{occ} \times s_{q,n}^{pres}) \qquad \text{(Generate foreground mask)} \qquad (21)$$

$$\mathbf{z}^{bg} \sim \text{ConvDRAW}_\theta^{bg}(\mathbf{r}) \qquad \text{(Sample background)} \qquad (22)$$

$$\hat{\mathbf{x}}_{bg}^q \leftarrow \text{BgRenderer}(\mathbf{z}^{bg}, \mathbf{v}^q) \qquad \text{(Decode background)} \qquad (23)$$

$$\hat{\mathbf{x}}^q \leftarrow \hat{\mathbf{x}}_{fg}^q + (1 - \alpha_{fg}^q) \times \hat{\mathbf{x}}_{bg}^q \qquad \text{(Render generations)} \qquad (24)$$

**Inference Process** The inference modules are paralleled with generative modules. We only highlight the different part below.

$$\mathbf{r}^q \leftarrow \sum_q f_{\text{repr-scene}}(\mathbf{x}^q, \mathbf{v}^q, \theta) \qquad \text{(Obtain scene-volume feature-map)} \quad (25)$$

$$\mathbf{z}_n^{pos} \sim \text{ConvDRAW}_{\theta,\phi}^{pos}(\mathbf{r}_n^q, \mathbf{r}_n) \qquad \text{(Sample 3D position for object } n) \quad (26)$$

$$z_n^{pres} \sim \text{CONV}_\phi(\mathbf{r}_n^q, \mathbf{r}_n) \qquad \text{(Sample global presence)} \quad (27)$$

$$\mathbf{s}_{q,n}^{pos} \leftarrow f_{3D \rightarrow 2D}(\mathbf{z}_n^{pos}, \mathbf{v}^q) \qquad \text{(Perspective projection)} \quad (28)$$

$$\mathbf{s}_{q,n}^{scale} \sim \text{ConvDRAW}_{\theta,\phi}^{scale}(\mathbf{r}_n, \mathbf{r}_n^q, \mathbf{s}_{q,n}^{pos}, \mathbf{v}^q) \qquad \text{(Sample object scale)} \quad (29)$$

$$\mathbf{x}_n^q \leftarrow \mathcal{ST}(\mathbf{x}^q, [\mathbf{s}_{q,n}^{pos}, \mathbf{s}_{q,n}^{scale}]) \qquad \text{(Crop object patch)} \quad (30)$$

$$\mathbf{r}_{q,n}^{what} \leftarrow \sum_q f_{\text{repr-obj}}(\mathbf{x}_n^q, \mathbf{v}^q) \qquad \text{(Encode object level context)} \quad (31)$$

$$\mathbf{z}_n^{what} \sim \text{ConvDRAW}_{\theta,\phi}^{what}(\mathbf{r}_n^{what}, \mathbf{r}_{q,n}^{what}) \qquad \text{(Sample global what)} \quad (32)$$

$$s_{q,n}^{pres} \sim \text{CONV}_\phi(\mathbf{r}_{q,n}^{what}, \mathbf{r}_n^{what}, \mathbf{s}_{q,n}^{pos}, \mathbf{s}_{q,n}^{scale}, \mathbf{v}^q) \qquad \text{(Sample local presence)} \quad (33)$$

$$\mathbf{z}^{bg} \sim \text{ConvDRAW}_{\theta,\phi}^{bg}(\mathbf{r}^q, \mathbf{r}) \qquad \text{(Sample background)} \quad (34)$$

## A.5 DATASET DETAILS

The sizes of objects are randomly chosen from $0.56$ to $0.66$ unit in the Mujoco physic world. Each object is put on the floor of 3D space with a range of $[-2, 2]$ along both x-axis and y-axis. We have three different types of object, cube, sphere, and cylinder with 6 different colors. For each dataset, we first randomly choose the number of objects in a scene, then randomly choose object type, color and their positions ($x$ and $y$ coordinates). For each scene, we have 30 cameras put at a radius of $3$, pointing at a squared area located at the center, thus, the camera would not always look at the center point. The pitch is randomly chosen from $[-\pi/6., -\pi/7.]$ and the yaw is randomly chosen from $[-\pi, \pi]$. During training, we randomly divide each scene into context set and query set, where the number of <image, viewpoint> pair in context set is randomly chosen from a range of $[10, 20]$ and the remaining are served as query set.

