# OpenReview forum: "OBJECT-ORIENTED REPRESENTATION OF 3D SCENES"
_ICLR.cc/2020/Conference — Reject_

### Official Review · AnonReviewer1 · 2019-10-24
**Official Blind Review #1**

**Rating:** 3

**Review:**

The paper proposes a model building off of the generative query network model that takes in as input multiple images, builds a model of the 3D scene, and renders it. This can be trained end to end. The insight of the method is that one can factor the underlying representation into different objects. The system is trained on scenes rendered in mujoco.

Summary of positives:
+The factoring makes sense, and the use of voxels + the physical property to enforce that two objects can't be superimposed in z_pres is a good strategy.
+There are a number of good qualitative results for understanding the learned object-oriented representation.
+The approach of learning 3D representations by comparing projections to observations is a good direction.

Summary of negatives:
- The method is quite complex and explained, in my view poorly (although I'm open to the other reviewers' opinion on the matter).
- The experiments are weak
- The manuscript makes fairly broad claims that aren't substantiated and ignores a great deal of work in the vision community on this topic.

Given the three largely orthogonal and fairly strong negatives, I lean heavily towards rejecting this paper. Independently each of these is an issue that would be push me to at least lean towards rejection. However, I would encourage a revision and resubmission with improved method explanation, stronger experiments, and a clearer picture with respect to existing work.

In more detail:

Method:
-I found the method section quite difficult to read, in part because the method is quite complex. This isn't intrinsically a bad thing, but complex methods with lots of steps should come with few surprises and descriptions that make the method accessible. In particular, the method section would benefit from a stronger figure that in part introduces the notation, as well as a little more thought in terms of the introduction of the method. A few instances:
1) "This is done by an order-invariant object encoder r_n = f_{order-invar-obj}(...)". One turns to the appendix, and tries to find this function. It's not explicitly there -- instead you need to match r_{n,C} = \sum_{i} .... , then look up above at the note that "ObjectRepNet is the module we use for object level order invariant encoder", then remember that sum is order invariant.
2) I searched throughout the paper and couldn't find precisely what model f3d->2d was. The figure suggests a projective camera and the text says "For more details on the coordinate projection, refer to Appendix.", but there's none in the appendix as far as I can see.
3) STN-INV is nowhere defined -- inverse spatial transformer?
4) s^{what} doesn't appear to be anywhere in the appendix -- is s^{what} factored into y^att and alpha^{att}? By matching the RHS, this seems to be the only possibility, but in the main body it's called ConvDRAW aka the GQN decoder, but in the appendix it's called Renderer.
5) There are lots of other little things -- e.g., a figure that refers to a variable that doesn't exist (see small details section )

I don't see why a paper this complex can't be accepted at ICLR, but I think at a minimum, the appendix should be more complete so that things are well-defined. I'm open to the possibility that I may just be slow so long as the other reviewers think the paper is crystal clear down to the details. However, I think even if I'm just the slow one, the authors should think about writing this more clearly and using consistent notation and function names.

Experiments:
-As far as I can see, the difference between ROOTS and GQN is that GQN is a little more blurry in its output (Figure 2) and ROOTS has a slightly better MSE for lots of objects (Table 1) but produces NLLs similar to GQN. There are a few issues with this:
(a) It's surprising that the correlation between larger numbers of objects and better MSE isn't really investigated -- why not show that GQN breaks at some point? The differences right now are fairly small, and I think the paper ought to delve into details to demonstrate that the differences are important.
(b) There are so many changes between ROOTS and GQN that I don't know if this has to do with the object-bits of it, or something else. This is part of a larger problem where there are no ablations. A large complex system is trained, and lots of changes are made to GQN. But when there are no ablations, it's unclear what parts of the changes are important and which parts aren't.
(c) It's not clear whether the GQN and ROOTS are being trained fairly -- do they have the same capacity? Why are they both trained for the same number of epochs? It seems entirely likely that ROOTS may train faster than GQN (or the reverse!). If there's only one experiment like this, why not train for a long enough time to ensure convergence and then take the checkpoint with best validation performance?
-The NLL results are very weak and probably not worth putting in, at least without some sort of explanation for why this gap is significant.
-The object detection quality experiment is incomplete -- I just do not know how to parse the numbers that are presented without some sort of simple baseline in order to make sense of things. Why not also try something like this on GQN?
-The qualitative experiments are nice but would be substantially improved by showing that:(a) that GQN doesn't do any of these (b) that ROOTs can train on 2-3 objects and test on 3-5 objects by simply changing the prior on K -- this is one of the primary advantages of object-centric representations of scenes (the ability to handle arbitrary numbers of objects).


Related work:

The paper really needs to make its claims more specific and position itself better with respect to related work.

Two gratuitous examples:
1) The title is "Object-oriented representation of 3D scenes", which covers decades of work in robotics and vision. This title should be changed.

2) "First unsupervised model that can identify objects in a 3D scene" is exceptionally broad: voxel segmentation is already a standard feature in point cloud libraries (e.g., "Voxel Cloud Connectivity Segmentation - Supervoxels for Point Clouds" Papon et al CVPR 2013). Is the manuscript and the Papon paper the same at all? No. But are they both unsupervised models that can identify objects in scenes. I'm not demanding that the authors write out a claim of novelty that's like a patent, but claiming "first unsupervised model that can identify objects in a 3D scene" is, in my opinion, clearly incorrect and needs to be qualified.


The paper should also better position itself compared to the wide variety of work that's been done on unsupervised 3D shape estimation/feature learning using reprojection. For instance (among many):
(1) Geometry-Aware Recurrent Neural Networks for Active Visual Recognition. Cheng et al. NeurIPS 2018
(2) Learning a multi-view stereo machine. Kar et al. NeurIPS 2017.
(3) Multi-view Supervision for Single-view Reconstruction via Differentiable Ray Consistency. Tulsiani et al. 2017.(4) Unsupervised Learning of Depth and Ego-Motion from Video. Zhou et al. CVPR 2017 (not voxels, but 2.5D or a form of 3D)

as well as the vast array of work on 3D reconstruction, including work that is object-oriented
(1) Learning to Exploit Stability for 3D Scene Parsing. Du et al. NeurIPS 2018.
(2) Cooperative Holistic Scene Understanding: Unifying3D Object, Layout, and Camera Pose Estimation. Huang et al. NeurIPS 2018
(3) Factoring Shape, Pose, and Layout from the 2D Image of a 3D Scene, Tulsiani et al. CVPR 2018
(4) Potentially not out at ICLR submission deadline, but 3D Scene Reconstruction with Multi-layer Depth and Epipolar Transformers. Shin et al. ICCV 2019.

I agree that there are differences between these works and the manuscript, but it's really peculiar to work on inferring a 3D volume of scenes from a 2D image or set of images, and only cite YOLO, faster RCNN, and FCNs from the world of CVPR/ICCV/ECCV etc where this work is done very frequently. These works do indeed sometimes rely on a bit more supervision (but not always). But they're tested on data that's far more complex than a set of spheres and cubes.



Small issues that do not affect my score.
- The claim that the method is unsupervised when it has access to precise camera poses seems a bit like a stretch to me. It's common enough that I've given up quibbling about it. Peoples' sense of distance is not exact. This deserves some further thought.
-The authors should go through and double check their use of GQN and CGQN -- it's said at the beginning that GQN just means CGQN, but then it's occasionally dropped (e.g., right before Table 1)
- Fig 1 shows z^{where}, which I guess got renamed?
- "The Scene Representation Network is modified bases on Representation Network in GQN." -> this sentence is presumably a typo/cut off halfway through.
- " This helps remove the sequential object processing as dealing with an object does not need to consider other objects if the features are already from spatially distant areas." -> this is unclear
- Eqn 11 appendix "sacle" -> "scale"
- "Object Representation Network" in A.2 "objcet" -> "object"
-Equation 15 -- the parentheses put i \in \mathcal{I}(D) inside the Renderer, which is presumably not true.
-Table 1 -- table captions go on top








----------------------------------------------------
Post-rebuttal Update:

AC: I would give a rating of 5 if the full revision is considered acceptable (since the paper is more clear), and increase to 4 if it is not (since there are some more experiments, although I think they are still quite weak).

I'm still inclined to reject the paper on the grounds of experimental comparisons and the open question of whether  but recognize that my concerns are ones which may not be shared by all communities (and that this is not my community).
1) Ablations/Comparisons to GQN: This may just be a cultural thing, but I'm puzzled by the claims that certain things (direct comparisons to GQN on feature representation, and ablations) don't need to be empirically done.

In my view, results really need to be empirically demonstrated. Simply stating that ROOTS has the capacity to decompose things into 3D and GQN doesn't have this built-in isn't enough. GQN has a feature vector, and it would not be surprising if it implicitly already did some of this. There have been far too many recent results in ML where a complex method is presented and it is asserted this complex method is necessary, followed by work that shows that  simple method does as well, typically due to issues in the dataset. I don't think expecting linear readouts of systems is too much of a burden to ask, if only to put the proposed work in context.

Indeed: the experiments in the revision show that ROOTS often does *worse* in generalization performance to previously unseen objects (improving in only 6/9). This is surprising -- if GQN is supposed to break, why doesn't it break here? I appreciate the author's response that there's a latent variable of # of objects that needs to be adjusted in the case of ROOTs, but this should be investigated.

The same thing goes with the claim that an ablation study is only necessary for improving results. This is just baffling -- is it possible that only certain parts of the method are necessary? Surely this is a problem that is worth studying. What if it's just that some aspect of the system has higher capacity than the equivalent in GQN and just works better?

AC: I realize that I'm just the cranky computer vision person shouting about numbers and I may be out of my element here, so take this as you want. But in my view, things really need to be evaluated (since vision struggled for many years because people showed a few nice qualitative results and didn't put their ideas to the test).

2) Clarity: I completely disagree with the authors that clarity issues were minor -- they really weren't and all reviewers agreed on this. Typos are thing lke tihs that are easy to read through without thinking. These were things that required thinking, flipping back-and-forth-etc.

The revision appears to be close to a full-rewrite, to the point where the diff system is useless -- it's all red/green. It appears to be clearer, but I haven't checked thoroughly. The ICLR 2020 guide is unclear how you should treat this (the AC guide says "you can ignore this revision if it is substantially different from the original version."). Personally, I don't think it's fair to authors who spent their time making their paper clear in the first place rather than on new results.



Smaller stuff:

-The authors have misunderstood my statement on paper complexity (although I now realize the comment has been edited)-- my point is that people who present a complex system have a strong obligation to present a clear explanation (since there's little opportunity for redundancy in explanation unlike a simple approach).

-"AP interpretable metric without a relative comparison": this is just not true although openreview is probably not the place to litigate this and I recognize that this is my outlook. Accuracy is highly interpretable: 90% top-1 accuracy on mnist would have been boring in 2005, and 90% top-1 accuracy on imagenet would be very exciting today.

-f_{3D-2D} There are multiple camera models. Skimming the revision suggests it's perspective projection, but the authors should realize that there are others (orthographic, weak perspective, etc) and they're often used because they're easier to learn with.

**Experience Assessment:**

I have published in this field for several years.

**Review Assessment: Checking Correctness Of Derivations And Theory:**

I assessed the sensibility of the derivations and theory.

**Review Assessment: Checking Correctness Of Experiments:**

I carefully checked the experiments.

**Review Assessment: Thoroughness In Paper Reading:**

I read the paper thoroughly.

---

> ### Author Response · Authors · 2019-11-10
> **Rebuttal to AnonReviewer1 - Part 1**
>
>
>
> We kindly suggest the reviewer first to check our general response: "For All Reviewers"
>
> We also have made some visualization here: https://sites.google.com/view/roots3d
>
> *Rebuttal Summary*
>
> > Thanks for the constructive review. We summarize our rebuttal here: (i) We agree that the description and Appendix need to improve by fixing typos, missing words, and minor errors. All will be thoroughly fixed in the revision. (ii) On experiments, we provide more clarification and reasoning in response to the questions. (iii) We agree on all points on related works and minor comments. We will add a significant amount of related works. We hope our argument sounds reasonable to the reviewer, and if so, we hope the reviewer to be open and flexible in adjusting the score. More detail discussions follow below:
>
> 1. Description clarity and model complexity.
>
> > We totally agree that those minor errors in our first submitted version could make readers feel that the method is complex. Thanks for pointing those in the description and Appendix. We will fix all pointed errors in the revision and will keep improving until the end of the rebuttal period, and further if accepted. We also would like to note that this problem of clarity should be separately considered from the inherent model complexity because the former can be fixed easily in revisions.
>
> > As the reviewer agrees, we believe the necessity of a more complex model should be considered in relation to the difficulty of the given problem. For example, it is clearly not a problem to see a model, designed to deal with complex natural videos, having a more complex architecture than a model designed for simple MNIST classification. Similarly, AIR (Eslami et. al. 2016), which is adding OO representation to VAE, has a significantly more complex model than VAE. SQAIR (Kosiorek, 2018), which is adding OO representation to Variational RNN, has also significantly more complex architecture than VRNN. We do not say that what they achieve is not interesting simply because they use a more complex model. These are reasonable architectures because they are solving a more complex new problem. (Of course, in the future we may see a simpler model for the same problem.) Our problem is also significantly more challenging and solves a very different aspect (obtaining OO representation) than the problem of GQN. [More is described in common answer A2].
>
> > We also believe that the reviewer’s argument makes sense if our claimed contribution is to improve a performance metric, e.g., generation quality or prediction accuracy, in comparison to other models (GQN in this case). In that case, we agree that getting better performance by merely using a more complex model may not be much surprising. But, this is not the case of our paper.
>
> Minor comments
> > We will fix all the pointed minor errors in the revision.
> - Minor 1 - missing explanation on order-invariant object encoder: We thought that the meaning would be clear for readers who are familiar with GQN. But, we will clarify it more in the revision.
> - Minor 2 - missing explanation on f_3D-to-2D: In Section 3.2, we mentioned that “using the 3D-to-2D coordinates projection operation f_{3D→2D}, we can compute the center location of an object existing in a context image.” In the revision, we will clarify it more and fix the missing section in the Appendix.
> - Minor 3 - missing explanation on STN-INV. Yes, inverse spatial transformer. We will clarify in the revision.
> - Minor 4 - s^{what}: We will clarify this in the revision.
> - Minor 5 - We will fix all in the revision.
>
> Reference.
> [1] SM Ali Eslami, Nicolas Heess, Theophane Weber, Yuval Tassa, David Szepesvari, and Geoffrey E Hinton. Attend, infer, repeat: Fast scene understanding with generative models. In Advances in Neural Information Processing Systems, pp. 3225–3233, 2016.
>
> [2] Kosiorek, Adam and Kim, Hyunjik and Teh, Yee Whye and Posner, Ingmar, Sequential attend, infer, repeat: Generative modelling of moving objects, Advances in Neural Information Processing Systems 2018.

---

> ### Author Response · Authors · 2019-11-10
> **Rebuttal to AnonReviewer1 - Part 2**
>
> [Continued from Rebuttal-Part-1]
> 2. Experiments.
>
> (a) “Why not show that GQN breaks in terms of MSE with a larger number of objects? The difference right now are fairly small.”
>
> >  We would like to refer the reviewer first to our common answer A1 and A5. We think the question may stem from a misunderstanding about our contribution and the purpose of Figure 2 and Table 1. The main contribution of this work is not to achieve better generation quality or MSE, but to obtain OO3D representation. Thus, the suggested experiment seems to be not very relevant to our claimed contribution because, unlike ROOTS, regardless of how many number of objects are given, GQN cannot provide object-level 3D representation.
>
> (b) Ablation study
>
> > Please find the corresponding answer in the answer to AnonReviewer2.
>
> (c) Fair training in terms of model capacity and training epoch.
>
> > Regarding the same capacity, we do not need to use the same capacity because we are not solving the same problem. The same argument we used before applies here: a video classification network does not need to use the same network capacity as an MNIST classification network. Similarly, AIR and SQAIR both use a significantly more complex network than their baselines. Regarding the training epoch, we trained both networks until they fully converge. We will clarify this in the revision.
>
> (d) NLL results are very weak.
>
> > As answered in A5 and A1 and (a) in the above, the purpose of this experiment is to show that our goal (OO3D representation learning) is achieved while not hurting other criteria (generation quality). So, we believe that the NLL results are still something worth to show that this purpose is satisfied. We will clarify this point more in the revision.
>
> (e) Why not try object-detection using GQN and compare to ROOTS
>
> > GQN cannot provide object detection because its encoding is scene-level, not object-level. So, without comparison to GQN, we think the number in Table 1 still provides some important baseline for follow-up research, and the precision-recall is still an interpretable metric even without a relative comparison.
>
> (f) Showing that GQN doesn’t do any of the qualitative tasks that ROOTS can do.
>
> > From the architecture of GQN, it is clear that GQN cannot do this because GQN only provides scene-level representation where objects are entangled. It's not about whether ROOTS can do it *better*, but GQN clearly cannot do it at all without a significant modification on the architecture or training objective. Please also read the common answer A2.
>
> (g) “The ability to handle an arbitrary number of objects” & “training on 2-3 objects and test on 3-5 objects by simply changing the prior on K“
>
> > “The ability to handle an arbitrary number of objects”: We think the question is a little vague to us, because “handle” could mean “generate”, “detect” or “composite” for ROOTS. We assume the reviewer is referring to the composition task. We give answer based on this assumption. This is an interesting suggestion. We think that ROOTS can composite a scene with a larger number of objects than the number of objects used in training scenes. This can be done by compositing learned object representations from other scenes. As stated previously, GQN cannot do this because it does not provide object-level representations. We will be running experiments for this and hope to add it in the revision if time allows.
>
> > “training on 2-3 objects and test on 3-5 objects by simply changing the prior on K“: We could not understand the part “changing the prior on K” and thus we answer it based on our best understanding of your question. We consider this as a generalization test on generation task w.r.t. the number of scene objects in training and test. We think both GQN and ROOTS will show some ability for this generalization but we are not sure if ROOTS should be better than GQN. The first scene encoding network of ROOTS seems to have the same capacity as GQN for this problem. But, we would like to note that this is not our claimed contribution (please refer to A1). This is not a weakness of the model, we simply have not considered this scalability factor in our design because that is not the goal of the model. Nevertheless, we think this is an interesting idea. We started running this experiment for our curiosity and hope to report the results within the rebuttal discussion time. Otherwise, we will consider adding it to the camera-ready.
>
> 3. Related Works and Others
>
> > Title - We agree. We will update the title in our revision.
> > “First unsupervised …” - We agree that the sentence is overly broad. We will update it in our revision.
> > Related Works. - We agree and thanks for pointing the related works. We agree that we need to cite more related works and clarify the position of our work in comparison to existing works. We will do it in our revision. Please also read the common answer A3 and A4.
>
> 4. Small Issues.
> Thanks for pointing these. We will fix the errors.

---

### Official Review · AnonReviewer2 · 2019-10-24
**Official Blind Review #2**

**Rating:** 3

**Review:**

TLDR: Interesting idea that seems promising, but lacks the maturity required to pass the ICLR bar: Lacks proper citations, comparisons with the latest works, no ablation study of their contributions.

The paper presents an extension of the Generative Query Network to incorporates 1) 3D grid for the state representations 2) hierarchical representation and 3) unsupervised model for explicit object representation (which is tied to 1.

The unsupervised representation is interesting, but this is a minor contribution on top of GCN and 3D representations have been widely studied in the vision community.

Also, except for the 3D representation, I am not sure how much the hierarchical representation helps. This leads to the question of why the authors did not perform ablation studies on each component.

Finally, it seems that the authors did not add proper citations. First, the 3D representation has been studied widely in the vision community and 3D-R2N2, ECCV'16 proposed using an RNN to encode a series of 2D images to learn 3D grid representation which seems quite similar to what the authors are proposing as an encoder and representation. Secondly, there are numerous methods on 3D neural rendering such as DeepVoxels, CVPR19 and all of the baselines in their experiments. The paper seems to completely ignore the works in this field.


Questions:

Please point out the equation number of implementation of Eq.2 in the appendix.

Is the y in Eq.3 the same y defined in the preliminary? Also, consider defining y again. There are almost two pages between the definition and Eq.3.


Minor:
Why learn the camera projection? $f_{3D\rightarrow 2D}$? Isn't this deterministic using a camera matrix?

Is the lighting fixed throughout the training and testing?

Sec.2 first sentence: an query --> a query

**Experience Assessment:**

I have published one or two papers in this area.

**Review Assessment: Checking Correctness Of Derivations And Theory:**

N/A

**Review Assessment: Checking Correctness Of Experiments:**

I assessed the sensibility of the experiments.

**Review Assessment: Thoroughness In Paper Reading:**

I read the paper at least twice and used my best judgement in assessing the paper.

---

> ### Author Response · Authors · 2019-11-09
> **Rebuttal to AnonReviewer2**
>
>
>
> We kindly suggest the reviewer to first check the general reply "For All Reviewers"
>
> We also have made some visualization here: https://sites.google.com/view/roots3d
>
> *Rebuttal Summary*
>
> > Thanks for the constructive review. We summarize our rebuttal here first: (1) We do not agree that our work is a minor contribution given GQN and literature in 3D computer vision. (2) We believe that an ablation study is not necessary in our case for the following reasons. First, the paper is not about improving a performance metric with a new architecture. Second, the result of some ablation study (e.g., removing hierarchical object representation) is obvious. (3) We agree that we need to add a significant amount of related works on 3D CV and we will do it in our revision. We hope our argument sounds reasonable to the reviewer, and if so we hope the reviewer to be open and flexible in adjusting the score. More detail discussions follow below:
>
> 1. Is our work a minor contribution on top of GQN and 3D representation studied in the vision community?
>
> > Please first refer to the related common answer A1 and A2 on why this is an important and challenging problem and the answer A3 on how our problem setting is different from existing literature in 3D computer vision. Given our arguments, if you still think it is a minor contribution, we would kindly like to ask answers from the reviewer on (1) how and why the existing works in 3D computer vision (specifically, which paper and what aspect of the model in the paper) make our contribution --- learning to disentangle each object in a scene containing many objects with occlusion and partial observability and learning its 3D representation --- a minor contribution, and (2) how our contribution can be considered a minor one, given GQN whose design is not intended for the object-level representation. We think these points were not justified in the first review.
>
> 2. Ablation study and how much the hierarchical representation help.
>
> > We would like to first refer to the related common answer A1, A2, and A5. We agree that an ablation study is needed when the goal of a paper is to improve some quantity like generation quality (in NLL or MSE) because then we need to identify which part contributes to the performance to what extent. But, this is not the case in our paper. The purpose of our new architecture is not to improve generation quality, but to obtain object-oriented representation from scenes with occluded multi-objects. The hierarchical modeling is required to *enable* (not to improve) the OO representation both in 3D and 2D-level. Thus, even without an ablation study, we believe its role is already clear from our existing experiment results, for example, from the fact that we obtain object-wise disentanglement in 2D images in Figure 3. When there is no hierarchical representation, the result is also clear without ablation study: we just cannot obtain such object-oriented representation at the 2D image level. And, in this case, we do not need to care about whether the performance decreases or not after removing the hierarchy. Furthermore, the generation quality after removing the hierarchy is not relevant to any of our claimed contributions. Also, similar papers like AIR and SQAIR that provide an OO representation version of VAE and VRNN, respectively, do not provide an ablation study but rather focus on what can be done with the OO-representation because that is what those papers are about.
>
> 3. Adding more related works from 3D computer vision.
>
> > We totally agree. Please refer to Answer A3 and A4.
>
> 4. Minor comments
>
> (a) Eqn number for Eq 2 in Appendix -> Will be updated in the revision.
> (b) Definition of y. -> yes, they are the same. It is introduced in page 2, and re-appears in the next page. We will remind the reader in the revision.
> (c) Why learn the camera projection? Isn't this deterministic using a camera matrix? -> The camera projection is a deterministic function and not learned. We will clarify this in the revision.
> (d) Is the lighting fixed throughout the training and testing? -> yes, the lighting is fixed throughout the training and testing.

---

### Official Review · AnonReviewer3 · 2019-10-28
**Official Blind Review #2400**

**Rating:** 3

**Review:**

The paper presents a framework for 3D representation learning from images of 2D scenes. The proposed architecture, which the authors call ROOTS (Representation of Object-Oriented Three-dimension Scenes), is based on the CGQN (Consistent Generative Query Networks) network. The paper provides 2 modifications. The representation is 1. factorized to differentiate objects and background and 2. hierarchical to first have a view point invariant 3D representation and then a view-point dependent 2D representation. Qualitative and qualitative experiments are performed using the MuJoCo physics simulator [1] (please add citation in the paper).

[1]Emanuel Todorov, Tom Erez, and Yuval Tassa. MuJoCo: A physics engine for model-based
control. In ICIRS, 2012.

+Learning 3D representations from 2D images is an important problem.
+The proposed methodology learns representations that are more interpretable, with higher compositionally.

While the paper takes a step towards a potentially impactful work, I cannot recommend it for publication in its current form.

1. There are claims in the paper that are not supported by the experiments. For example, “As seen in Figure 2, ROOTS has clearer generations than GQN. ” However, Figure 2 does not show this at all. It shows no difference between ROOTS and GQN.

2. The paper can benefit from further clarity throughout—in general it seems a bit rushed. For example, the caption on Figure 3 reads “For example, GoodNet segments a scene into foreground and background first and decompose foreground into each individual object further…” The text has not discussed what GoodNet is. Its also unclear what is depicted in each of the columns in Figure 3. This should be clearly explained.

3. I suggest clarifying Figure 1 further and referring to it in section 3. Its currently not referred in the text although it is an overview of the proposed architecture.


Other comments
-Table 2: Why not have a precision-recall curve (as is standard) and report average precision numbers?
-Table 2: Why not compare to CGQN?

Minor
-There are typos throughout the text.

**Experience Assessment:**

I have published in this field for several years.

**Review Assessment: Checking Correctness Of Derivations And Theory:**

I assessed the sensibility of the derivations and theory.

**Review Assessment: Checking Correctness Of Experiments:**

I carefully checked the experiments.

**Review Assessment: Thoroughness In Paper Reading:**

I read the paper at least twice and used my best judgement in assessing the paper.

---

> ### Author Response · Authors · 2019-11-09
> **Rebuttal to AnonReviewer3**
>
>
>
> We kindly suggest the reviewer please first check another reply "For All Reviewers"
>
> We also have made some visualization here: https://sites.google.com/view/roots3d
>
> *Rebuttal Summary*
>
> > Thanks for the review. In this review, we could not find an argument pointing to the major limitation of our paper. We found what is pointed are either (i) a misunderstanding of the reviewer and hasty generalization, or (ii) minor things like typos or missing reference that can be fixed easily. It is quite surprising to see that: first, the reviewer uses these minor errors that can be very easily fixed, as the major factors of the decision, while another reviewer (R1) points these as constructive minor comments for further improvements not affecting the score. Second, there is no question or discussion about the main contribution (see A1) of the paper. We hope our argument sounds reasonable to the reviewer, and if so, we hope the reviewer to be open and flexible in adjusting the score. More detail discussions follow below:
>
> 1. The reviewer claims that the paper contains claims that are not supported by the experiments, and mention that we have a sentence “ROOTS has clearer generations than GQN” while the reviewer cannot see a difference *at all*.
>
> > Please first refer to the answer A1 and A5 in the above common answers. To summarize, first, the purpose of the pointed experiment is to show that our goal (OO3D representation learning) is achieved while not hurting other criteria (generation quality). So, the generation quality of ROOTS doesn’t need to be better than that of GQN. Second, we definitely see a difference where ROOTS generates sharper edges that are closer to those in the ground-truth images while GQN generates a bit more blurry images in general. This point is also confirmed by another reviewer AnonReviewer 1 describing in his/her review that “GQN is a little more blurry (than ROOTS) in Figure 2”. So, although it can be seen not a significant difference, we believe it is not correct to generalize and say that we are claiming what is not supported **at all** by the experiments. Please, clarify if there are other such arguments not supported by the experiments. In the revision, we will update the sentence more to clarify how and why they are different.
>
> > For this kind of minor comments, reviewers usually suggest moderating the tone of the sentence.
>
> 2. The reviewer says the text has not discussed what GoodNet is in the caption of Figure 3. It’s also unclear what is depicted in each of the columns in Figure 3”
>
> > Thanks for pointing this. GoodNet was the name of the model before we changed it to ROOTS.  In our revision, we will fix the typo in the caption. We will also clarify the meaning of the columns in Fig. 3. We believe that these are minor factors that can be easily fixed and thus usually suggested as a minor comment not affecting the score.
>
> 3. Figure 1 is not referred to in the text.
>
> > Thanks for pointing this. We will update it in the revision. We believe that this is also a minor factor that can be easily fixed and thus usually suggested as a minor comment not affecting the score.
>
> 4. Minor comment
>
> 4.a. Making precision recall-curve table to a curve & comparison to CGQN
>
> > We already provide the precision-recall table. We think it is also a good idea to make it a precision-recall curve.
>
> 4.b. Why not compare to CGQN?
>
> > Our baseline is indeed CGQN. So, it is compared to CGQN. In the submission version, we already mentioned that, for brevity, we use ‘GQN’ in our paper to actually mean CGQN. In the revision, we will clarify again at the beginning of the Experiment section, that the ‘GQN’ label actually means CGQN.

---

### Official Review · AnonReviewer4 · 2019-11-13
**Official Blind Review #4**

**Rating:** 6

**Review:**

Overview:
This paper is certainly very interesting, unlike other papers and makes some very solid contributions. The qualitative results are very impressive. Unfortunately, the paper is very poorly written. If the authors can address the issues below and improve the quality of the writing, I would recommend that this paper be accepted. However, in its current state, I recommend that this paper be rejected.

Major:
The claim that “This is the first unsupervised model that can identify objects in a 3D scene” is not true. There is MONet and Iodine that can identify objects without supervision in 2D projections of 3D scenes. This claim should be revised.

What is r_C in p(z|c, r_C)?

The authors are using non-standard GQN notation and their notion is not consistent. The authors should make their notation consistent or use the standard GQN notation. Section two would not make sense to people that are not already familiar with GQN.

The scene volume map is interesting, the inductive bias preventing two objects being present in the same location is interesting, however one could imagine a case where uncertainty in the model could lead to two objects being represented in the same cell.

The terms in Equation 3 should be explained more explicitly. The way I understand it p(s_n|z, x) is the view-point dependent representations of the objects and this is why you condition on z, x. Placing p(s_n|z, x) in the text where you talk about s_n being view dependant with some explanation would help.

Why do you use a Gaussian distribution for the position? Would it not make more sense to use a uniform distribution? Could this be related to bias in your data? I.e. more objects in the centre of the scene?

What happens if object n is not present in the context image, y_c? In this case what is s_{c,n}^pos. Also, this notation: r_n = f({y_c,n}c)  is a little ambiguous. I assume it means that you are applying the function f to the set of all patches in y_c? Also, what is the object invariant object encoder? A reference to details in the appendix or a footnote would suffice.

It’s great that you are able to exploit the coordinate transform and use it for rendering.

The results are very impressive: being able to swap in and out objects in a scene, showing the 2D renderings of single objects from different view-points and the scene decompositions and predicting where the “missing” object is (Figure 6).

Minor:
The introduction could be strengthened with additional references. There are claims that object-wise factorisation will help with transfer, it would be good to have references to other work that supports this view. Also the claim that humans have 3D representations for objects requires a reference.

Typos (there are too many to list here, these are just a few):
* Abstract: and and rendering
* “and”s should be replaced with commas in the second line of the intro.
* Generally the paper is not written well.
* The GQN, as a conditional → is a conditional
* Target observations (in section 2) does not need a capital.
* This sentence does not make sense: “instead of encoding compressing the whole scene”
* Because of intractable posterior

There are many additional grammatical errors.


-----------
Edit: Following changes made to the paper, I am now more satisfied. The writing should still be improved further and suggest that the authors fully revise the paper before the camera ready version, if the paper is accepted. I have increased my score to 6.

**Experience Assessment:**

I have published one or two papers in this area.

**Review Assessment: Checking Correctness Of Derivations And Theory:**

I assessed the sensibility of the derivations and theory.

**Review Assessment: Checking Correctness Of Experiments:**

I carefully checked the experiments.

**Review Assessment: Thoroughness In Paper Reading:**

I read the paper thoroughly.

---

> ### Author Response · Authors · 2019-11-15
> **Thanks for the constructive review!**
>
> We are deeply grateful for the constructive review. We found all of the comments are reasonable. We also would like to thank the reviewer for acknowledging many of our contributions and the importance of the problem/model/demonstration. Above all, we totally agree with the readability problem of the submitted version of the paper. Thanks to the reviewers, we are significantly rewriting the paper focusing on the purpose of improving readability. We will provide a much more readable and consistent version of the paper in our revision.
>
> We also have made some visualization available here: https://sites.google.com/view/roots3d
>
> In the following, we respond to other questions
>
> * “This is the first unsupervised model that can identify objects in a 3D scene” is not true.
>
> > Yes, we totally agree. We will remove or more clarify that sentence in our revision
>
> * What is r_C in p(z|c, r_C)?
>
> > r_C is the embedding of the scene C. It is the output of scene invariant encoder. In the revision, the meaning will be clear.
>
> * Inconsistent or non-standard GQN notation
>
> > In the revision, we will do both. We will use GQN notation and make it consistent.
>
> * Section 2 would not make sense to people that are not already familiar with GQN.
>
> > We will also clarify Section 2 so that such people can also understand GQN easily.
>
> * Uncertainty in the model could lead to two objects being represented in the same cell.
>
> > Thanks for pointing this out. This is an interesting question. In our model, this situation is fundamentally prevented by design because a cell can only represent one object. We agree that when observing a limited context, there will be high uncertainty leading to many possible explanations. In this case, our design of the scene-volume map is to encourage the model to find explanations where two objects are not existing in the same position.
>
> * The terms in Eqn 3 should be explained more explicitly. ... you talk about s_n being view-dependent with some explanation would help.
>
> > Thanks for this suggestion. We totally agree and will apply this in our revision
>
> * Gaussian distribution for the position? Why not uniform?
>
> > The mean and variance of the Gaussian distribution are learned and thus it is not to represent a preference to a center area of the scene. We want to model the uncertainty of the position, ideally giving a high probability around the actual position while having a low probability for areas far from the actual position. We think the bell-shape of the Gaussian distribution seems proper to model this.
>
> * What happens if object n is not present in the context image, y_c? In this case what is s_{c,n}^pos. Also, this notation: r_n = f({y_c,n}c)  is a little ambiguous. I assume it means that you are applying the function f to the set of all patches in y_c? Also, what is the object invariant object encoder? A reference to details in the appendix or a footnote would suffice.
>
> > With the camara-coordination projection function f_{3D->2D}, we can know that the position after the mapping is not inside the context image y_c. Thus, in this case, we do not crop any patch from that context image. We will update the notation clearly. Yes, your understanding on the function is correct. We will also clarify what object invariant encoder either in the main text or in the Appendix.
>
> * It’s great that you are able to exploit the coordinate transform and use it for rendering. The results are very impressive: being able to swap in and out objects in a scene, showing the 2D renderings of single objects from different view-points and the scene decompositions and predicting where the “missing” object is (Figure 6).
>
> > Thanks again for acknowledging our contribution
>
> * Minor
> - More reference in the Introduction: We agree. We will add relevant references.
> - Typos: Thanks for pointing these. We will fix all of them in our revision.

---

> ### Author Response · Authors · 2019-11-15
> **Thanks for the re-evaluation!**
>
> We are very grateful for re-evaluating the paper and adjusting the score. In the revision, we believe that we indeed substantially rewrote many parts of the paper, particularly for the clearer exposition of the technical description. We hope you to enjoy the revised version. Thanks.

---

### Author Response · Authors · 2019-11-09
**For ALL Reviewers - Part 1/2**

We thank all the reviewers for taking time read our paper and provide insightful feedback. We would like to first provide answers and further clarification commonly applying to all reviewers. So, we kindly suggest all reviewers read this part first. We use the term GQN as a general framework including both the original GQN and CGQN. We first upload this rebuttal while working on the revision. What we promise here will all be updated in the coming revision. We use OO to stand for Object-Oriented and OO3D for OO and 3D.

[A1] Our main contribution is not to improve 3D generation quality.

> but is to learn, in the GQN setting, to obtain disentangled OO3D representation, as written in Introduction. This representation is independent, modular and 3D-viewpoint-invariant. GQN cannot provide such representation due to its scene-level representation where objects are entangled. We believe that obtaining such representation is a very important problem in deep representation learning, which is the main theme of this conference. The main purpose of obtaining such representation is not to improve generation quality but to enable new important abilities such as compositionality, transferability, better generalization, and variable binding (for causal inference and reasoning), which are main unsolved challenges in contemporary deep learning. Thus, our experiments focus on demonstrating such advantages of OO3D representation (compositionality and transferability). Therefore, based on our understanding, the main decision criteria should be based on the claimed contribution: (i) the importance of making it possible to obtain such representation in a challenging setting and (ii) how well the benefits of the representation (e.g., compositionality and transferability, etc.) is demonstrated.

[A2] We achieve this in a significantly challenging setting.

> (i) Our model is unsupervised, not using any annotations on voxels, cloud points, meshes, segmentation or object box. (ii) It is generative and learns both representation and rendering. (iii) It learns a single 3D-representation from which multiple views can be generated. (iv) It is end-to-end. (v) It deals with scenes with multiple objects with occlusion and partial observability. Solving these problems altogether is a significantly challenging problem and has not been tried before. As explained below [A3], some existing works from 3D computer vision are somewhat relevant to a part of the above challenges, but to our knowledge, we are not aware of any work that deals with the same level of challenge (particularly, (iii) and (v) are rare in 3D CV literature). Also, the GQN design does not consider object-level representation, and thus, although we start from GQN, still a significant amount of investigation, observation, ideation, design and optimization is required to develop a new model that can achieve the new abilities.

[A3] Relation to 3D computer vision (CV).

> The problem setting of those works in 3D CV is quite different from ours. To our knowledge, they focus on either (i) supervised approaches using voxel, cloud points, or mesh annotations, or (ii) generating images of different 2D perspectives of a 3D scene (e.g., using GANs) but not obtaining the 3D-representation, i.e., there is no single representation from which multiple views can be generated, or (iii) solving single object problems rather than disentangling objects from a scene containing multiple objects.

[A4] Related Works.

> We totally agree that we should discuss the relevant works from 3D computer vision in our initial submission. We thank reviewers for pointing this. We will discuss the pointed papers and also others we found relevant. We did not intend to ignore those important works which we also have been inspired by.

[A5] The goal of the experiments in Fig 2 and Table 1 (on qualitative generation quality, NLL and MSE, in comparison to GQN) is to show that our goal (learning OO3D representation) is achieved while not hurting other criteria (generation quality).

> As discussed in [A1], our goal is not to show that the proposed model can significantly improve generation quality. We would like to note that it is not easy to retain this quality under the constraint of the discrete representation structure in ROOTS. Specifically, using discrete structure in neural networks provides advantages such as interpretability and compositionality but usually comes with some performance degradation. This is because it limits the model space and optimization compared to continuous representation. In our case, we however actually achieve a comparable generation quality that is less blurry than that of GQN (as agreed by AnonReviewer #1). Thus, we think the argument ---our model achieves minor contributions because of little improvement in generation quality--- is not correct. Instead, the representation aspect of the proposed model and its importance should be considered as the main criteria for the score.

---

> ### Author Response · Authors · 2019-11-15
> **For All Reviewers - Part 2/2**
>
> We agree that the paper should be improved with the notation error, inconsistency, and readability in general. We are significantly changing the writing of the paper focusing on this purpose of readability.

---

### Author Response · Authors · 2019-11-15
**About Revision [to All Reviewers]**

We are deeply grateful to all reviewers for taking the time to read our paper and providing constructive reviews. Following the reviewers' comments, we have made a significant update in the uploaded revision. While solving minor errors and clarifying points asked by the reviewers, we have focused on the following major updates in the revision.

First of all, we significantly improved the readability of the paper. For this, we have not only fixed the notation error and inconsistency but almost fully rewrote and reorganized a substantial amount of parts in the paper. We adopted the standard GQN notation (as suggested by reviewer 4) and made all notations consistent throughout the paper including the Appendix. We fixed the missing descriptions on notations. We made Section 2 more complete so that a reader who is not familiar with GQN can understand better. We believe, with all the above updates, the revised paper is much more readable and shows the contribution more clearly.

The second major focus of the update is to add the description positioning our proposed work in the existing literature of 3D computer vision. We totally agree that we missed this important discussion in our initial submission and appreciate the reviewers for pointing this. We introduced a paragraph in Section 1 to clarify the different settings between our proposed work and existing works. Also, we added a considerable amount of discussion in the Related Works section along with relevant citations. While we have tried our best for this positioning and reference citation, we are willing to improve it even further if reviewers suggest additional reference.

Finally, we added two more experiment results in the Appendix, following reviewer1's suggestion. In the first experiment A.1, we evaluate the generalization performance in the settings where the number of objects in training scenes is different from that in test scenes. In the second experiment A.2, we evaluate the composition generalization. In this setting, we compose a scene with 9 objects by using object components trained from scenes with only up to 3 objects. We show that ROOTS still generates high-quality images with proper occlusion handling.

With the revision, the paper is 9 pages long. We hope the reviewers kindly consider the fact that our paper should contain much larger images for figures in the experiments than other papers to report our results properly.

Along with the above major improvements, we believe that we have addressed all concerns of each reviewer and look forward to a positive reconsideration on the paper and hearing feedback about the updated version. We hope the reviewers can take our responses and revisions into consideration when evaluating our final score. Thanks again for your reviews!

---

### Decision · Program_Chairs · 2019-12-19

**Decision:**

Reject

**Comment:**

The author proposes a object-oriented probabilistic generative model of 3D scenes.  The model is based on the GQN with the key innovation being that there is a separate 3D representation per object (vs a single one for the entire scene).  A scene-volume map is used to prevent two objects from occupying the same space. The authors show that using this model, it's possible to learn the scene representation in an unsupervised manner (without the 3D ground truth).

The submission has received relatively low scores with one weak accept and 3 weak rejects.  All reviewers found the initial submission to be unclear and poorly written (with 1 reject and 3 weak rejects initially).  The initial submission also failed to acknowledge prior work on object based representations in the 3D vision community.  Based on the reviewer feedback, the authors greatly improved the paper by reworking the notation and the description of the model, and included a discussion of related work from 3D vision.  Overall, the exposition of the paper was substantially improved.  Some of the reviewers recognize the improvement, and lifted their scores.

However, the work still have some issues:
1. The experimental section is still weak
The reviewers (especially those from an computer vision background) questioned the lack of baseline comparisons and ablation studies, which the authors (in their rebuttal) felt to be unnecessary. It is this AC's opinion that comparisons against alternatives and ablations is critical for scientific rigor, and high quality work aims not to just propose new models, but also to demonstrate via experimental analysis how the model compares to previous models, and what parts of the model is necessary, coming up with new metrics, baselines, and evaluation when needed.

It is the AC's opinion that the authors should attempt to compare against other methods/baselines when appropriate.  For instance, perhaps it would make sense to compare the proposed model against IODINE and MONet.  Upon closer examination of the experimental results, the AC also finds that the description of the object detection quality to be not very precise.  Is the evaluation in 2D or 3D?  The filtering of predictions that are too far away from any ground truth also seems unscientific.

2. The objects and arrangements considered in this paper is very simplistic.

3. The writing is still poor and need improvement.
The paper needs an editing pass as the paper was substantially rewritten.  There are still grammar/typos, and unresolved references to Table ?? (page 8,9).


After considering the author responses and the reviewer feedback, the AC believe this work shows great promise but still need improvement.  The authors have tackled a challenging and exciting problem, and have provided a very interesting model.  The work can be strengthened by improving the experiments, analysis, and the writing.  The AC recommend the authors further iterate on the paper and resubmit.  As the revised paper was significantly different from the initial submission, an additional review cycle will also help ensure that the revised paper is properly fully evaluated.  The current reviewers are to be commended for taking the time and effort to look over the revision.